# South Pacific sea surface temperature and global ocean circulation changes since the late Miocene

Antje Wegwerth [1] ✉, Helge W. Arz [1], Jérôme Kaiser [1], Gisela Winckler [2], Lester Lembke-Jene [3], Vincent Rigalleau [3], Nicoletta Ruggieri[3], Henrik Sadatzki [3,4] & Frank Lamy [3]

The Antarctic Circumpolar Current (ACC) is a major driver of global ocean circulation and climate. To better understand the interplay between long-term atmospheric and ocean variability in the Southern Ocean since the late Miocene, we present sea surface temperature (SST) and carbonate preservation records from the Subantarctic Eastern South Pacific (IODP Site U1543), along with an extended ACC strength record from Central South Pacific Site U1541. We focus on long-term eccentricity-scale variations showing decreased (increased) SST with enhanced (reduced) $CaCO_3$ preservation, and stronger (weaker) ACC strength, particularly during the Pliocene. These changes coincide with stronger (weaker) South Pacific SST gradients, possible northward (southward) migration of Southern Ocean fronts, strengthened (weakened) westerlies, and atmospheric $CO_2$ release. These patterns contrast with Pleistocene glacial-interglacial cycles. Reduced Pacific-Atlantic exchange through the Drake Passage may have weakened Atlantic Meridional Overturning Circulation during warming at Site U1543 across the intensification of Northern Hemisphere Glaciation. Simultaneous stronger ACC and higher $CaCO_3$ deposition in the high-latitude Pacific suggest a strengthened basin-wide Pacific overturning circulation during parts of the Pliocene.

Earth's climate system experienced important temperature changes since the late Miocene, which was expressed by long-term cooling of up to 15 °C in northern and southern high latitudes, but interrupted by episodic warmings[1,2]. The Meridional Overturning Circulation (MOC) distributes heat, salt and carbon across all ocean basins and, thus, plays a central role in global climate change[3]. A key component of the MOC is the Antarctic Circumpolar Current (ACC) in the Southern Ocean (SO) with a surface to deep water circumpolar flow that likely evolved during the late Miocene cooling ~10 Ma[4] approximating its modern circulation characteristics in the South Pacific ~5 Ma[5] (Fig. 1). The ACC, the strongest zonal current system, circles Antarctica and links all

three oceans affecting to a vast extent ocean heat transport and global climate variability[3,6]. Influenced by the extent of the Antarctic Ice Sheet (AIS) and sea ice, the intensity of the ACC depends on the strength of the Southern Westerly Winds (SWW) and the large-scale density structure of the SO[3,6,7]. Both affect the position of the frontal system and are controlled by changes in equator-to-pole temperature gradients[7]. Today, pronounced latitudinal SST gradients characterise the SO from the Subtropical to the Polar Front[8]. The northern and southern boundaries of the ACC are in the north of the Subantarctic Front and in the south of the Southern ACC Front, respectively[9]. As the position of the individual oceanic fronts is highly variable in the

[1]Marine Geology, Leibniz Institute for Baltic Sea Research Warnemünde (IOW), Rostock, Germany. [2]Lamont-Doherty Earth Observatory, Columbia University, Palisades, NY, USA. [3]Helmholtz Center for Polar and Marine Research, Alfred Wegener Institute, Bremerhaven, Germany. [4]MARUM-Center for Marine Environmental Sciences, University of Bremen, Bremen, Germany. ✉e-mail: antje.wegwerth@io-warnemuende.de

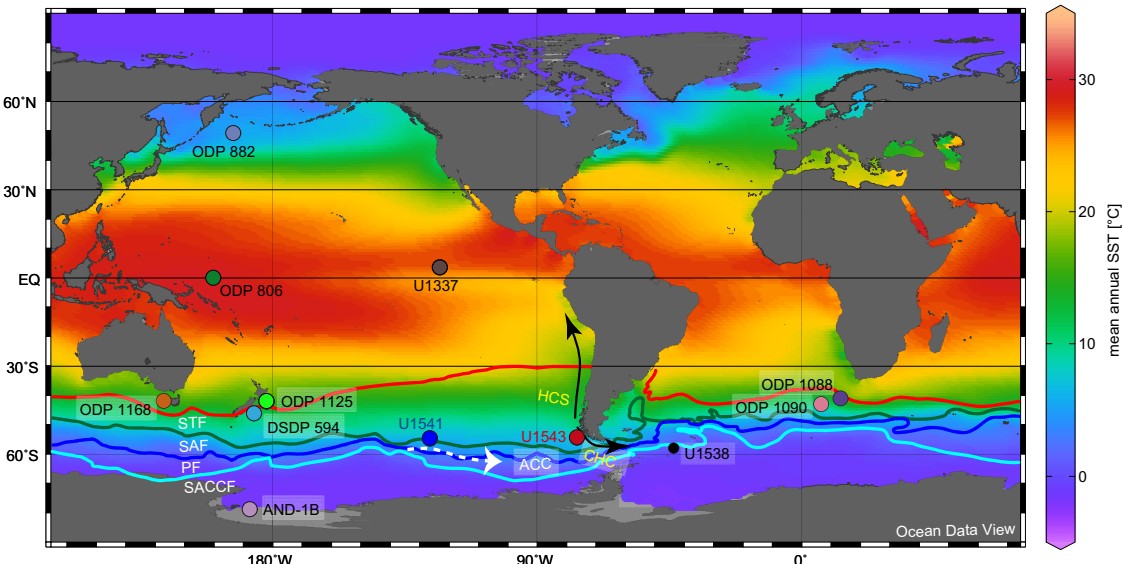

**Fig. 1 | Global sea surface temperature and selected study sites.** Map with mean annual sea surface temperatures (SST) showing the location of IODP Expedition 383 sites U1543 in the Eastern South Pacific Ocean, U1541 in the Central South Pacific Ocean and other sites discussed in the text. HCS Humboldt Current system, CHC Cape Horn Current, ACC Antarctic Circumpolar Current, STF Subtropical Front, SAF Subantarctic Front, PF Polar Front, SACCF Southern ACC Front. Map created with Ocean Data View[71].

different parts of the SO, depending e.g. on bathymetric features (Fig. 1), the ACC differs in its latitudinal extent as well[9,10].

The development of the modern ACC was likely linked to the intensification of the North Atlantic deep-water (NADW) and Ross Sea bottom water production coinciding with the starting Panama seaway closure around 9–10 Ma[5]. The latter's final closure around 5 Ma is suggested to be responsible for further intensification of the Atlantic MOC (AMOC) and the establishment of the modern ACC system[5]. The position of the frontal system of the ACC varied through the past. For instance, there is evidence that the ACC migrated by up to 2° equatorward and up to 6° poleward over the last two deglaciations as reconstructed by sea surface temperature gradients in the southern Indian Ocean[11]. A detailed knowledge of ocean circulation changes associated with shifts in SST gradients, front migrations and ACC strength is lacking for most of the Neogene. Recently, Lamy et al.[10] reconstructed the Pliocene to Pleistocene ACC strength in the Subantarctic Central South Pacific (CSP; Site U1541; Fig. 1) for the last 5.3 Myrs and documented pronounced 400,000-year eccentricity cycles (Supplementary fig. 2). Strongest ACC intensities were reported during the globally warmer-than-present periods of the Pliocene and showed consistent glacial-interglacial variations during the past ~1 Myrs[10]. The same pattern with an enhanced Cape Horn Current flow as part of the ACC system during interglacials is reported for the Subantarctic Eastern South Pacific (ESP) at the entrance of the Drake Passage (Site U1542) for the last 800 kyrs[12]. However, SST records beyond the Pleistocene are sparse in the South Pacific[1,13,14] and a more complete understanding of the dynamics between South Pacific temperature variability and the ACC dynamics regarding its strength, latitudinal position and extent is still missing. Changes of the atmospheric temperature around Antarctica are known to directly affect changes in the ocean circulation and atmospheric $CO_2$ during glacial-interglacial cycles, which is called the positive feedback loop[15]. This means that warming triggers a strengthening ocean circulation and rise of atmospheric $CO_2$, which in turn further amplifies warming[15]. Therefore, Neogene SO temperature records are important for better understanding the interplay among climate, ocean circulation and $CO_2$ changes, and to improve climate models and more accurately project the consequences of future global warming. Additionally, regional SST

gradients help understanding past changes in the dynamics of the oceanic fronts of the SO, interactions between high and low latitudes, and exchange of carbon between ocean and atmosphere[7,16,17]. Shifts in meridional SST gradients in the SO can strongly affect the strength and position of the SWW and, therefore, the strength and position of the ACC[7,16], while pronounced temperature contrasts characterise frontal boundaries[7,10,18,19].

Here, we investigated sediment cores drilled during International Ocean Discovery Programme (IODP) Expedition 383 in the Subantarctic ESP and the CSP (Fig. 1; Site U1543: 54°35.06′S, 76°40.59′W, 3860 m water depth; Site U1541: 54° 12.756′ S, 125° 25.540′ W, 3604 m water depth[20]). In the ESP (Site U1543), alkenone-based sea surface temperature estimates ($SST_{UK'37}$) were used to detect the timing and amplitude of warming and cooling in the ESP during the last 8 Myrs and to reconstruct meridional SST gradients in the Pacific Ocean by using additional $SST_{UK'37}$[21–23] records from the Pacific Ocean. We chose Site U1543 because its location in the Subantarctic ESP in close vicinity to the Drake Passage reflects water exchange along the cold-water route driven by the ACC, and latitudinal frontal migrations linked to temperature gradients and SWW strength[24,25]. Today, the SST at that site is about 7 °C (Fig. 1).

To estimate past changes in ACC intensity and in its latitudinal position in relation to SST gradients, the sedimentary Zr/Rb ratio was used. In the SO, Zr/Rb ratios are known to be linearly correlated with grain size as evidenced by established sortable silt-flow speed correlations[10,12,25–27]. Here, we extended the ACC strength proxy record from the CSP (Site U1541), formerly covering the last 5.3 Myrs[10], back to 8 Ma, i.e. the late Miocene, with similar methods used previously[10]. While the ACC intensity in the CSP reflects deep flow conditions at the East Pacific Rise, SST gradients represent surface ocean features sensitive to atmospheric temperature, wind stress and oceanic frontal positions. Therefore, integrating ACC strength and SST gradients provides valuable insights for estimating general changes in South Pacific Ocean circulation. We are aware that the ACC strength record is from the CSP (Site 1541) and thus not at the same site as the SST record. SST changes at the ESP Site 1543 might be additionally affected by the proximity to the continent and the complex interaction of the Subantarctic ACC with the Cape Horn Current and thus not necessarily

reflect the mean atmosphere-ocean circulation in the Pacific ACC upstream[12].

For complementing investigations of the positive feedback loop[15], involving the tight interplay between temperature, ocean circulation and deep ocean $(CO_3)^{2-}$, we compared both the $SST_{UK'37}$ at ESP Site U1543 and the ACC record at CSP Site U1541 to the carbonate preservation record at Site U1543 based on changes in $CaCO_3$ content. The latter reflects a combination of surface primary productivity and deep-water carbonate preservation[28]. The preservation and dissolution of carbonate in the Pacific Ocean strongly relate to deep water mixing and ventilation[29–35]. A stronger deep water mixing and ventilation can cause upwelling of corrosive, $(CO_3)^{2-}$-rich deep water to the surface, thereby releasing $CO_2$ to the atmosphere. As the deep water progressively releases $CO_2$, the potential for $CaCO_3$ preservation increases. Therefore, the sedimentary $CaCO_3$ content can indirectly provide information on past carbon exchange between the ocean and the atmosphere. In the deep ESP, dissolution intensity is apparently the main factor triggering late Pleistocene glacial-interglacial carbonate variability, which is supported by a comparison of sedimentary carbonate with foraminiferal fragmentation ratios and shell weights, with the latter two independent of productivity[35].

## Results and discussion

### Temperature variability in the South Pacific Ocean during the last 8 Myrs

The ESP $SST_{UK'37}$ record ranges between 3 °C and 16 °C and largely reflects the global cooling trend from the late Miocene towards the Pleistocene[1,2,18,19,23,36–39] (Figs. 2, 3). We note that the present $SST_{UK'37}$ record represents the surface ocean conditions on a 25-kyr resolution, hence reflecting general long-term trends. Such resolution cannot reflect glacial-interglacial or millennial-scale cycles. Hence, amplitudes of the $SST_{UK'37}$ variability might be underestimated, which, in turn, could have an effect on the long-term trends. Nevertheless, $SST_{UK'37}$ amplitudes are relatively high (-13 °C), which seems typical for that region as previously shown for glacial cycles during the late Pleistocene[12,40]. Possible shifts in alkenone-producing assemblages most likely not biased the $U^{K'}_{37}$ paleothermometer due to the positive linear correlation between the $U^{K'}_{37}$ and the $U^{K'}_{38Me}$ indices (methods, Supplementary figs. 3, 4).

The main long-term characteristics of the SST development in the ESP (Fig. 2d) are the late Miocene cooling (7.8–6.0 Ma; ΔSST: −9 °C), the warming during the Pliocene global temperature reversal[1,41] at the Miocene-Pliocene transition (6.0–5.2 Ma; ΔSST: 7 °C), the long-term cooling during the first half of the Pliocene (5.1–3.7 Ma, ΔSST: −9 °C), the long-term warming trend during the second half of the Pliocene towards the Early Pleistocene (3.0–2.3 Ma; ΔSST: 7 °C) and the long-term Pleistocene cooling (since 2.1 Ma, ΔSST: −7 °C).

Although Site U1543 $SST_{UK'37}$ record mirrors the global climate trend with cooling towards the Pleistocene, there are two particular intervals deviating from other temperature records[1,2,37]. The first phase is the pronounced mid-Pliocene cooling of about 5 °C around 3.7 Ma that lasted for about 200 kyrs and marks the coldest episode during the Pliocene in the ESP record (Fig. 2d). $SST_{UK'37}$ decreased to 5 °C and was 2 °C lower than today in this region. This cooling seems to be a prominent feature in the ESP since it is less pronounced in the global benthic δ[18]O record[36] or in other Southern and Northern Hemisphere $SST_{UK'37}$ records[1,2,37], where SST decreased by no more than 2 °C (Figs. 2, 3). Nevertheless, similar strong cooling may have occurred in the SW Pacific and offshore Tasmania, for which both coolings are represented, however, only by one data point and are not the coldest period of the Pliocene, respectively[1,18] (Fig. 3f, g). Similarly, this cooling is not evident in a stacked temperature record from the southern hemisphere extratropics[2] (30–45°S; Fig. 2c), which indicates that the SO is not necessarily cooling or warming uniformly.

The second and most noticeable pattern of the Site U1543 $SST_{UK'37}$ record is the long-term and pronounced warming that parallels the intensification of the Northern Hemisphere Glaciation[42,43] (iNHG; 2.7–2.4 Ma[44]). This warming lasted for at least 700 kyrs (from 3.0 to 2.3 Ma) and $SST_{UK'37}$ rose to 12 °C reaching values comparable to the period before the onset of the late Miocene cooling (Fig. 2d). The ESP $SST_{UK'37}$ record, therefore, documents long-term Pleistocene cooling not before -2.1 Ma. This timing is consistent with southern hemisphere extratropical cooling around 1.8 Ma[2]. These temperature shifts in the ESP, which partly differ from previous climate reconstructions, are most likely related to regional changes in the Pacific affecting the ACC and associated oceanic frontal dynamics.

### Changes in the South Pacific Ocean circulation since the late Miocene

The reconstructed ACC strength in the CSP[10] shows a similar long-term pattern as seen for the $SST_{UK'37}$ variability in the ESP particularly during the Pliocene and early Pleistocene (Fig. 4a, b; Supplementary fig. 2). Contrary to the behaviour during warm interglacials of the last 790 kyrs[12], the ACC generally strengthened during the longer-term cooling periods and weakened during longer-term warming periods during the Pliocene and early Pleistocene. For instance, during the prominent cooling phase around 3.7 Ma, the reconstructed ACC strength increased up to 160% relative to the Holocene, while it weakened to 60% during the iNHG ESP warming[10] (Fig. 4b).

We reconstructed meridional SST gradients between the eastern equatorial Pacific[23] and the Subantarctic ESP (U1337–U1543) from 100 kyr-binned SST records (Figs. 1, 4d). A possible relation between SST gradients and ACC intensity seems reasonable since the basin-spanning SST gradient reflects the atmospheric status over the South Pacific, including the SWW. The SST gradients are enhanced during long-term ACC maxima of the Pliocene and early Pleistocene until ca. 2 Ma (Fig. 4b, d). During that time, conditions were not as warm as during the late Miocene and not as cold as during the middle or late Pleistocene, hence representing a long-term intermediate climate state (Fig. 4a).

The increased latitudinal SST gradients in the ESP were possibly associated with waxing of the AIS that caused strengthened SWW[1,10] and enhanced not only the ACC strength, but also moved the position of oceanic fronts and the ACC northward (Fig. 4). Periods with Pliocene ACC maxima show reduced SST up to about 5 °C in the ESP. Such temperatures occur today between the Subantarctic and the Antarctic Polar Front (APF)[8] (Fig. 1), which supports a northward ACC shift. Nevertheless, we assume an APF position still south of ESP site U1543, considering the lower-than-late Pleistocene AIS volume[45,46] and generally warmer SST during the Pliocene and early Pleistocene (Fig. 4a, f). The observed relation between stronger ACC and colder conditions on the long-term timescale is less expressed during the late Miocene, which might be related to a generally smaller expansion of Antarctic sea ice as well as weaker[10] and more southward oceanic fronts due to still warmer than Pliocene conditions until ~6.5 Ma.

With regards to the entire investigated period, the general strengthening ACC since the late Miocene[4] was potentially also triggered by lower-latitude cooling and increased SST gradients between Southern Hemisphere middle to low latitudes as shown in the Tasmanian gateway area[18,19,23,38] (Fig. 3g, j, k). It was recently proposed that the equatorial to subtropical SST gradient might have had the strongest impact on SO fronts at least in the Tasmanian gateway region[18,19]. While SST gradients between the equator and the STF increased since the late Miocene cooling, SST gradients apparently decreased between the STF and higher latitudes in the Tasmanian gateway area[18,19]. This difference between the SST gradients was explained by stronger cooling in the STF region when compared to relatively stable Antarctic SST[18]. The Subantarctic ESP SST record clearly documents pronounced

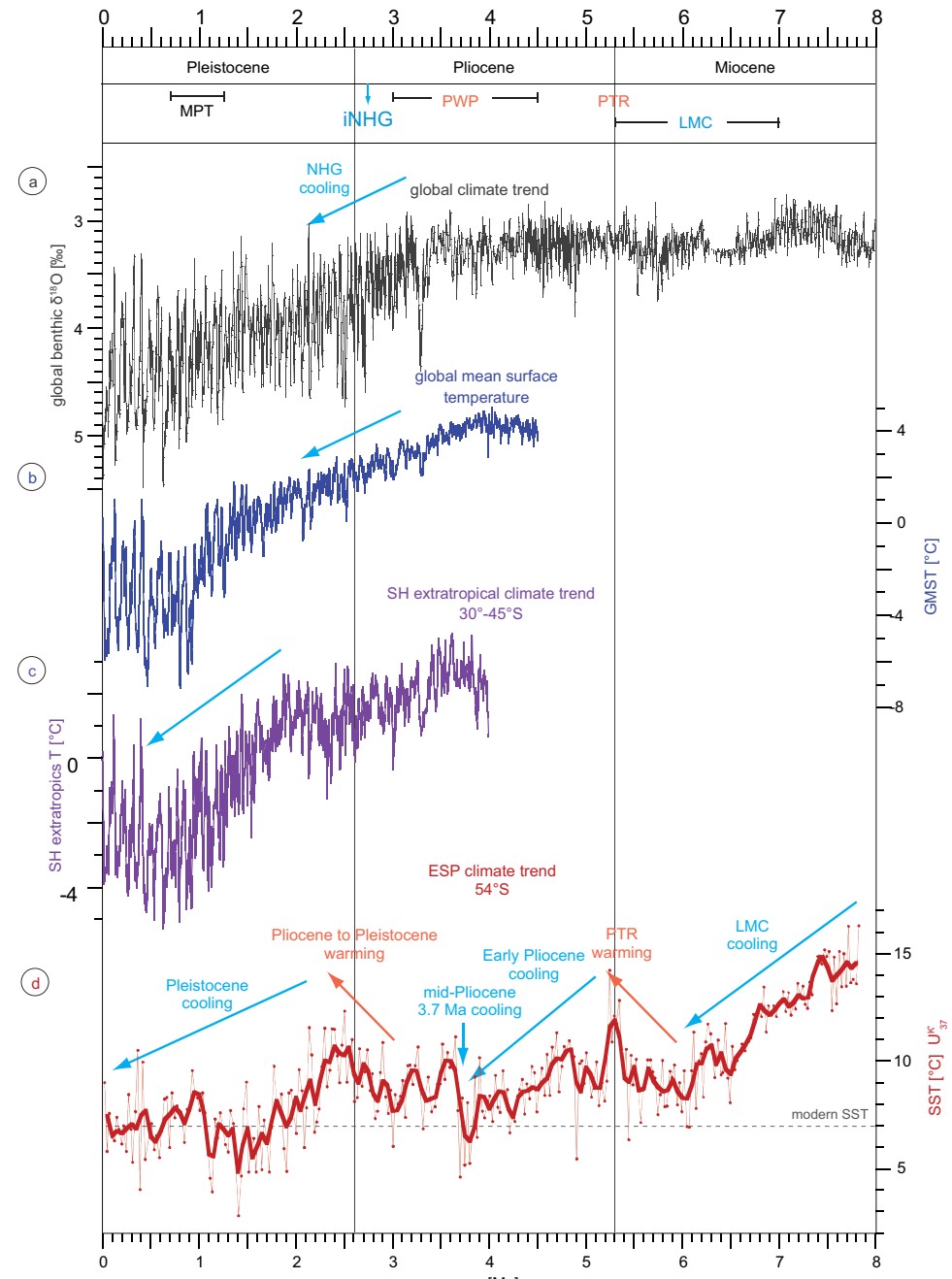

**Fig. 2 | Global, Southern Hemisphere and Eastern South Pacific temperature variability since the late Miocene.** Temporal variation of **a** the global ocean benthic $\delta^{18}O$ reflecting global temperature[36], **b** global mean surface temperature difference from today (GMST)[2], **c** temperature difference from today in Southern Hemisphere extratropics[2] and **d** the alkenone-based sea surface temperature in the Eastern South Pacific (ESP) Ocean (SST$_{UK'37}$; U1543/PS97-114-2; this study). The thick smoothed SST$_{UK'37}$ record represents a 100 kyr-binned record by averaging the SST$_{UK'37}$ data for every 100 kyrs in 50 kyr-overlapping windows. LMC late Miocene cooling, PTR Pliocene global temperature reversal, PWP Pliocene warm period, iNHG intensification of the Northern Hemisphere Glaciation, MPT mid-Pleistocene transition.

cooling at higher southern latitudes and generally increasing equator-to-pole SST gradients since the late Miocene on a multi-million-year timescale. The apparent discrepancy between SST gradients of the equator to the Subantarctic ESP (increasing) and of the STF to high latitudes (decreasing) is not necessarily contradicting, but rather reflecting both the pronounced southern high-latitude cooling and the corresponding northward frontal migrations. The increasing equator-to-pole SST gradient from the ESP represents the overall cooling since the late Miocene with stronger SST decrease at higher southern latitudes compared to the equatorial realm. This is in accordance with a

strong high-latitude impact on meridional SST gradients (polar amplification) in both hemispheres[47].

## Pacific Ocean circulation changes and the carbonate preservation effect

The reconstructed ACC intensity in the CSP shows a remarkable resemblance to the CaCO$_3$ preservation record in the ESP from the Pliocene to the late Pleistocene. Particularly during intervals of maximum ACC speed, exceeding the Holocene strength by up to 160%, CaCO$_3$ preservation from the ESP increased considerably with CaCO$_3$

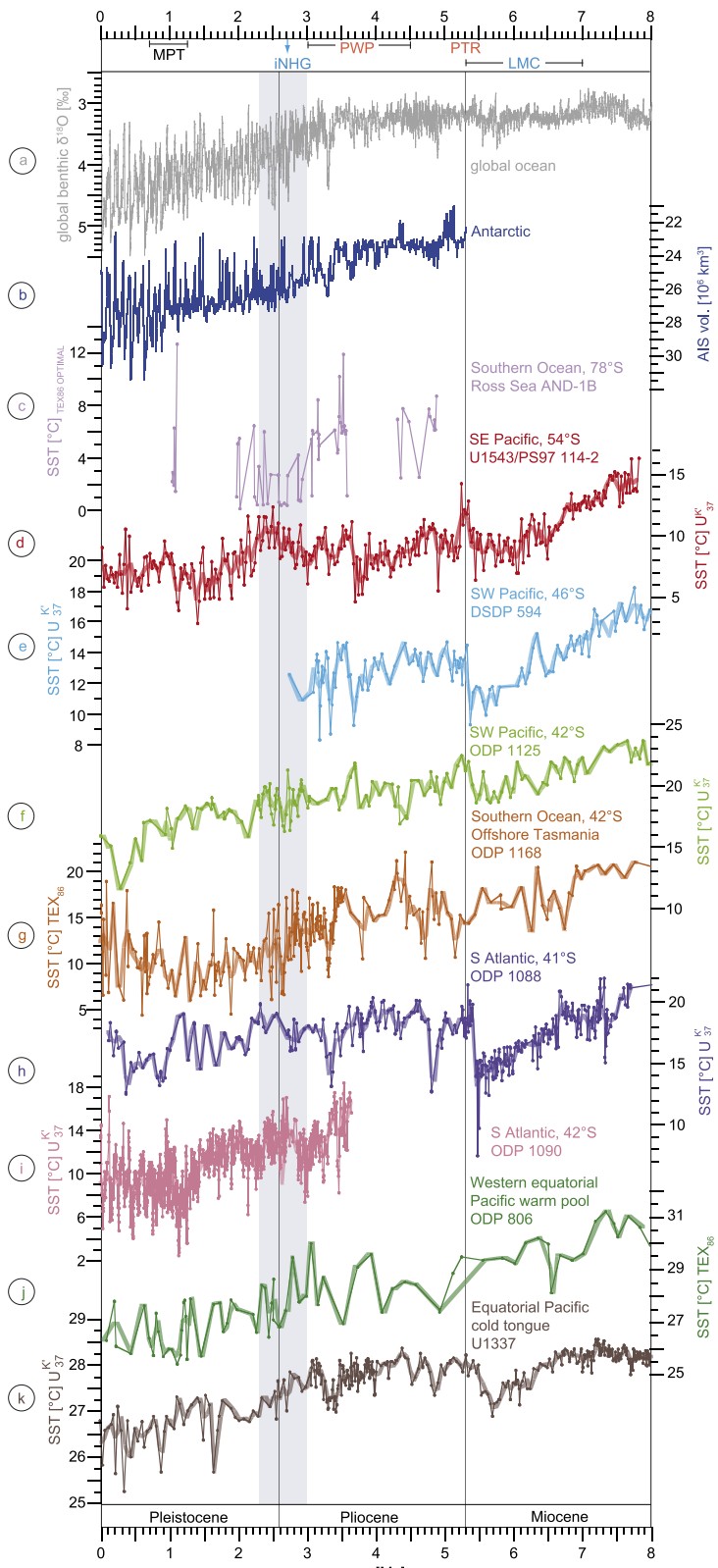

**Fig. 3 | Global and southern hemisphere climate variations since the late Miocene. a** global benthic oxygen isotope signature[36]; **b** simulated volume of the Antarctic Ice Sheet[46] (AIS; reversed scale); **c–k** sea surface temperature (TEX_86 and SST_UK'37) records from, respectively, the Ross Sea (AND-1B[39]), the Eastern South Pacific Ocean (U1543/PS97-114-2; this study), the Southwest Pacific Ocean[1] (DSDP 594, ODP 1125), the Tasmanian gateway area (ODP 1168[18]), the South Atlantic (ODP 1088[1] and ODP 1090[37]), the western equatorial Pacific warm pool (ODP 806[38]) and the equatorial Pacific cold tongue region (U1337[23]). The thick smoothed records in **d–k** represent 100 kyr-binned records by averaging the SST data for every 100 kyrs in 50 kyr-overlapping windows. Grey bar during iNHG marks the period of ESP warming. LMC late Miocene cooling, PTR Pliocene global temperature reversal, PWP Pliocene warm period, iNHG intensification of the Northern Hemisphere Glaciation, MPT mid-Pleistocene transition.

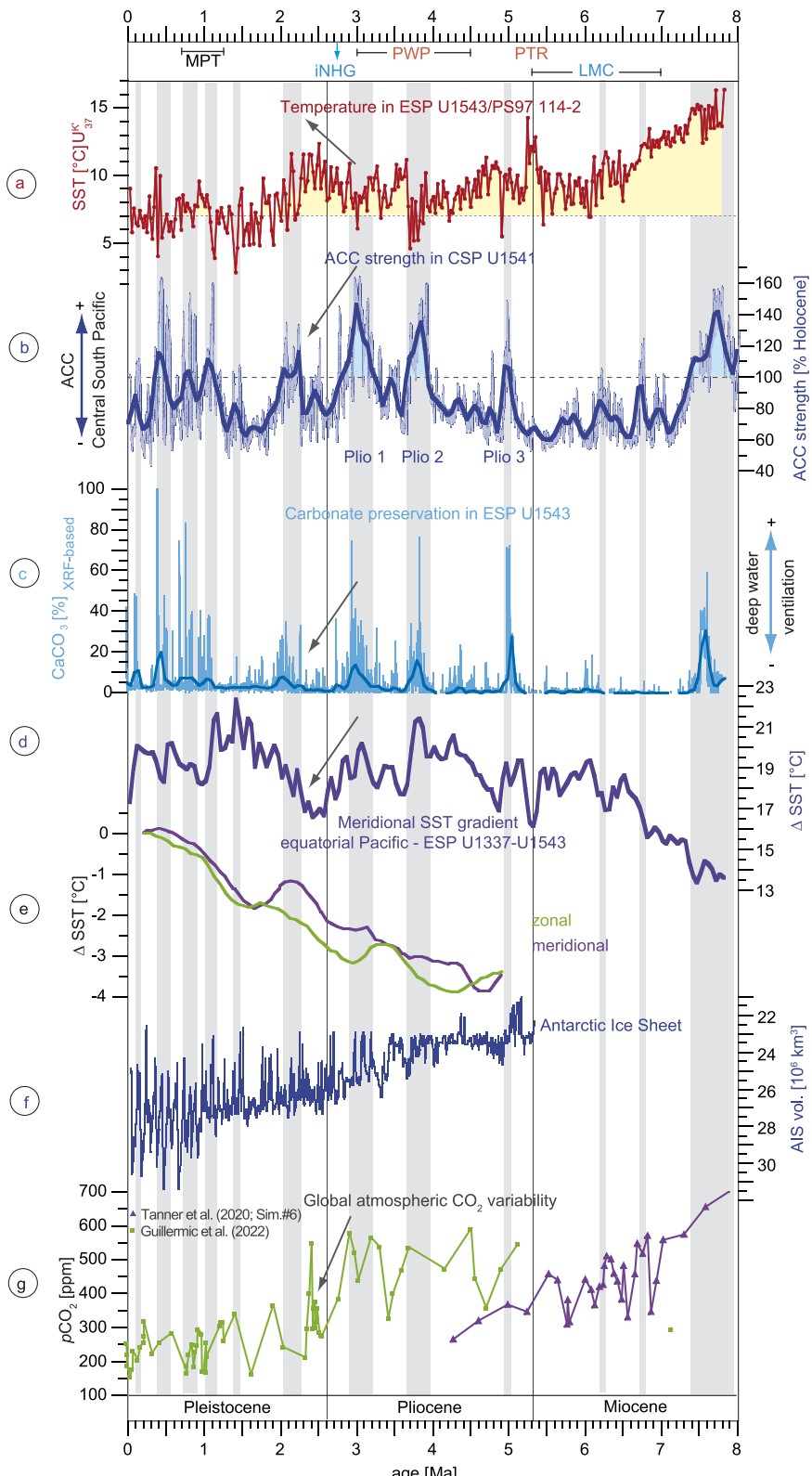

**Fig. 4 | Changes in South Pacific temperature and ocean circulation over the last eight million years.** Temporal variation of **a** alkenone-based sea surface temperatures in the Eastern South Pacific Ocean (SST$_{UK'37}$; U1543/PS97-114-2; this study; yellow shading denotes warmer than modern conditions), **b** strength of the Antarctic Circumpolar Current (ACC) in the Central South Pacific (CSP) Ocean based on Zr/Rb ratios of site U1541 (Lamy et al.[10]; 5.3-8 Ma this study), **c** CaCO$_3$ content at site U1543 based on XRF representing changes in deep water ventilation, **d** meridional SST gradients between the equatorial Pacific[23] (ODP1337) and the

Eastern South Pacific Ocean (U1543, this study), **e** computed global zonal and meridional SST gradients[16], **f** simulated volume of the Antarctic Ice Sheet[46] (AIS; reversed scale) and **g** reconstructed changes in global atmospheric $pCO_2$[63,65]. The thick smoothed records in **b**, **c** represent 100 kyr-binned records by averaging the SST data for every 100 kyrs in 50 kyr-overlapping windows. LMC late Miocene cooling, PTR Pliocene global temperature reversal, PWP Pliocene warm period, iNHG intensification of the Northern Hemisphere Glaciation, MPT mid-Pleistocene transition. Grey vertical bars denote periods of intensified ACC strength.

contents of up to ~80% (Fig. 4b, c), which suggests a tight relation between the ACC system and the deep water dynamics in the ESP.

Today, ESP Site U1543 is located in a water depth of 3860 m and influenced by lower circumpolar deep water (lower-CDW), whereas in the past it was potentially also affected by Antarctic Bottom Water (AABW)[20,25,35]. A pronounced glacial-interglacial variability of $CaCO_3$ preservation at Site U1543 is documented for the last 140 kyrs[35], with higher $CaCO_3$ preservation during interglacials and low to no carbonate preservation during glacials[35], a pattern that is similar to carbonate records in the southern Indian Ocean, South Atlantic, and western South Pacific[35]. Due to the expansion of Antarctic sea ice contributing to AABW formation, it is proposed that during glacials, more corrosive lower-CDW reached ESP Site U1543, being ultimately responsible for $CaCO_3$ dissolution and less preservation, respectively[25,35]. A reduced deeper water ventilation and enhanced uptake of $CO_2$ in the deeper water could have additionally contributed to the decreased carbonate preservation during glacials since ACC intensity was relatively low[10,12,25].

The enhanced $CaCO_3$ preservation at Site U1543 occurred during the warm late Pleistocene interglacials[35] and, counterintuitively, also during the longer-term cold periods of the Pliocene and early Pleistocene. However, in both cases, the ACC intensity strengthened considerably[10,12]. A more vigorous deep ocean circulation, reflected here by the ACC strength, can result in mixing and ventilating the water column and hence release $CO_2$ from the deep water to the atmosphere, which facilitates carbonate preservation in the deep ocean[29–35]. Therefore, we propose that the Pliocene to early Pleistocene long-term periods of enhanced $CaCO_3$ preservation could reflect intervals of increased ventilation of the deep ESP water favoured by the strengthened ACC. Since conditions were generally warmer during the Pliocene and early Pleistocene, (Fig. 4a), AABW formation and the influence of $CO_2$-rich AABW and/or lower CDW was possibly lower when compared to the late Pleistocene glacials.

The relation between ACC and $CaCO_3$ is less clear during the late Miocene, possibly due to much weaker meridional SST gradients that may have caused an ACC strength mainly not exceeding the Holocene level (Fig. 4d). Minimum SST during the late Miocene long-term cooling were generally at least 2 °C higher than today, which is in most parts of the modern SO more similar to SST at the STF. Therefore, the weaker SST gradients and warmer conditions during the late Miocene, together with weaker and more southward oceanic fronts, were likely responsible for weaker SWW and ACC strength in the Subantarctic South Pacific. Although the configuration of the AIS extent is unclear for the late Miocene, the warmer conditions were likely related to a lower than Pliocene-to-Pleistocene AIS extent. This could have been associated with a larger latitudinal extent of the ACC, explaining the lower ACC strength. Less carbonate preservation during Miocene periods with enhanced but generally lower than Holocene ACC strength additionally points to less deep ocean ventilation and less $CO_2$ outgassing hampering carbonate preservation. The strongly decreasing global atmospheric $CO_2$ during the late Miocene cooling[1] might be related to the reduced carbonate preservation.

Periods with a stronger ACC strength particularly during the mid to late Pliocene occurred on average every 400 kyrs[10] (Fig. 4b, Supplementary fig. 2). Thus, not only a higher than Holocene ACC strength, but also a relatively long duration of enhanced ACC likely improved long-term mixing of the water column and deep water ventilation, which progressively released $CO_2$ to the atmosphere. The impact of the ACC on atmospheric $CO_2$ during the 400-kyr eccentricity cycles from the Pliocene to the early Pleistocene was previously suggested, based on the covariation with the global stacked benthic $\delta^{13}C$ record[10,36], supporting an important role of the SO for atmospheric $CO_2$ variability.

A sediment record from the subarctic Western North Pacific Ocean (Figs. 1, 5; ODP Site 882; 50°N) covering the last 5.9 Myrs

presents similar long-term $CaCO_3$ and SST gradient patterns as in the ESP. It has to be noted that the reconstructions of the SST gradients are based on 100 kyr-binned $SST_{UK'37}$ records, hence cannot accurately reflect conditions during glacial-interglacial cycles and should be viewed as long-term trends. This also explains why the records of SST gradients may not completely capture the much higher resolved $CaCO_3$ records, but reflect similar long-term changes. The long-term periods of enhanced $CaCO_3$ contents in the North and South Pacific during the Pliocene may indicate a concurrent basin-wide or bipolar deep water ventilation triggered by strong meridional SST gradients in both hemispheres[22,34]. Burls et al.[34] related the deep water ventilation and preservation of carbonate in the North Pacific Ocean to a strengthened Pacific Meridional Ocean Circulation (PMOC) either established by radiative forcing mechanisms[34] and/or salinity changes[48]. The concomitant higher $CaCO_3$ contents in the high-latitude South and North Pacific Oceans could, therefore, support a strengthened basin-wide PMOC that coincided with a strengthened ACC. Model simulations have shown that under Pliocene boundary conditions, an activation of PMOC is associated with a strengthened Drake Passage throughflow[49], which is ultimately controlled by the ACC strength. Therefore, an intensified PMOC might not only go along with an enhanced Drake Passage throughflow, but also a strengthened ACC at least in the Pacific. Thus, since North Pacific deep water upwells in the SO[50] and ACC intensities and Pacific convection were concurrently enhanced during parts of the Pliocene[10,34], we suggest a relation between PMOC and ACC on the scale of Pliocene 400 kyr eccentricity cycles[10], similarly to the possible dependence of an enhanced AMOC on the ACC[5]. More data are needed to investigate the potential relation and the mechanistic behaviour between Pacific deep water circulation and the ACC during the past. At the current state, we can only suppose a mutual role of the North and South Pacific circulation (i.e. meridional and circumpolar) in modulating atmospheric $CO_2$ changes during the Pliocene in a long-term perspective.

## South Pacific Ocean warming during the intensification of the Northern Hemisphere Glaciation

A pronounced and long-term warming occurred in the ESP during the iNHG (Fig. 4a). $SST_{UK'37}$ strongly augmented from ~7 to 12 °C between 3.0 and 2.3 Ma, hence contrasting cooling in the northern hemisphere[1]. This SST increase is largely consistent with SO warming and Antarctic sea ice retreat during this time interval as indicated by proxy data and modelling results[1,2,13,37,51–53]. Although interrupted by a cooling, the SST record from the Ross Sea[39] shows warmings of up to 6 °C at 2.4 Ma and 2.2 Ma (Fig. 3c). Similarly, general warm conditions occurred in the SW Pacific[1] and Tasman Sea[18] with a cooling not before 2.4 Ma (Fig. 3f, g). Even though punctuated by short cooling periods as well, two SST records from the South Atlantic[1,37] reflect warming around 2.6 Ma and 2.4 Ma (Fig. 3h, i). The high-resolution stacked extratropical Southern Hemisphere temperature record suggests intermittent warming of at least 2.5 °C around 2.5 Ma between 30°S and 45°S[2] (Fig. 2c). Foraminiferal assemblages from sediments of the Ross Sea continental shelf indicate warm super interglacials at 2.95 Ma, 2.31 Ma and 2.0–1.83 Ma with reduced sea ice cover[53]. Similarly, diatom and opal analyses suggest that conditions were warmer and sea ice was reduced along the Antarctic Peninsula between 3.1 and 2.2 Ma[54]. Simulations have indicated a partial reduction of the Antarctic ice volume during the period from 2.7 to 2.4 Ma[45,46] (Fig. 4f). A record of ice-rafted debris from the SW Atlantic (Fig. 1; U1538) suggests multiple fluctuations of the Western AIS during the Pliocene-Pleistocene transition, indicating a highly dynamic and unstable AIS during the iNHG[52]. Therefore, the long-term warming during the iNHG not only in the ESP but also in other parts of the SO was possibly associated with episodic AIS melting and southward migration of oceanic fronts.

The resulting decrease in the eastern equatorial Pacific to Subantarctic ESP SST gradient (Fig. 4d) for that period explains the long-

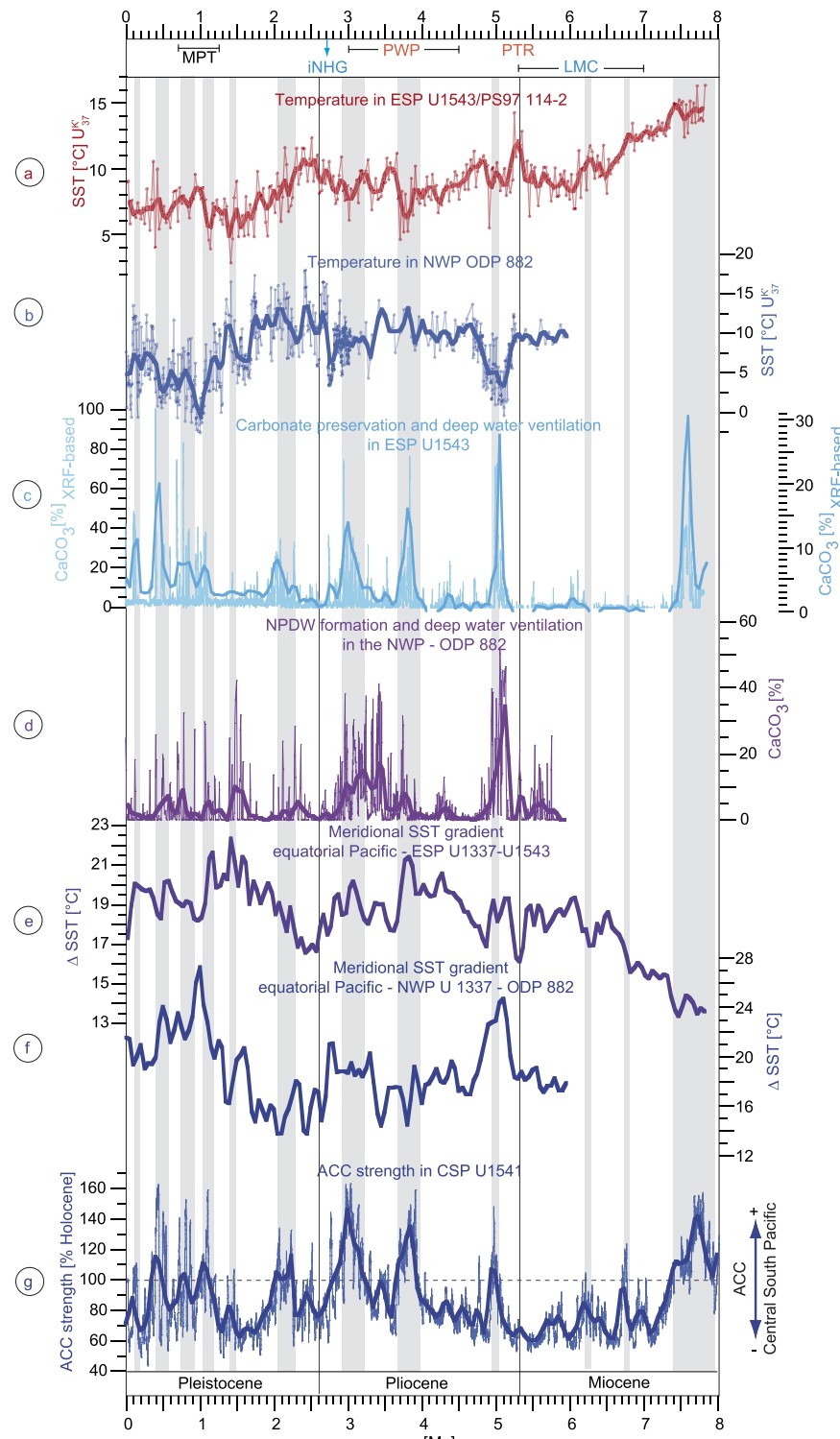

**Fig. 5 | South and North Pacific changes in temperature and ocean circulation since the late Miocene.** Temporal variation of alkenone-based sea surface temperatures (SST_UK'37) in **a** the Eastern South Pacific Ocean (U1543/PS97-114-2; this study) and **b** in the Northwest Pacific Ocean[22] (ODP 882), **c** CaCO₃ content at site U1543 based on XRF representing changes in deep water ventilation, **d** CaCO₃ content in the North Pacific Ocean[34] (ODP 882) representing changes in deep water ventilation, **e** meridional SST gradients between the eastern equatorial Pacific Ocean[23] (ODP1337) and the Eastern South Pacific Ocean (U1543, this study), **f** meridional SST gradients between the eastern equatorial Pacific Ocean[23]

(ODP1337) and the NW Pacific Ocean[22] (ODP 882) and **g** strength of the Antarctic Circumpolar Current (ACC) in the Central South Pacific (CSP) Ocean based on Zr/Rb ratios of site U1541 (Lamy et al.[10]; 5.3–8 Ma this study). The thick smoothed records represent 100 kyr-binned records by averaging the proxy data for every 100 kyrs in 50 kyr-overlapping windows. LMC late Miocene cooling, PTR Pliocene global temperature reversal, PWP Pliocene warm period, iNHG intensification of the Northern Hemisphere Glaciation, MPT mid-Pleistocene transition. Grey vertical bars denote periods of intensified ACC strength.

term weakening of the ACC and decreased carbonate preservation in the South Pacific (Fig. 4b, c). Such reduced meridional SST gradient during the iNHG was not seen in a previous study (Fig. 4e), where gradients were computed[16] by using low- and mid-latitude paleodata from the Pacific and Atlantic Oceans. Nevertheless, high-latitude SO SST data, where warming is evidenced[1,2,13,37], are not implemented in that reconstruction, which is most likely the cause for the apparent discrepancy.

It is possible that meltwater derived from the AIS could have changed the density structure of the SO in regions close to the AIS. This, in addition to the reduced meridional SST gradient in the ESP and potentially in other regions of the SO, could have affected the ACC and thus the global MOC. The impact of Antarctic meltwater on the SO and global MOC is manifold[55–61]. Enhanced freshwater supply into the SO results in surface water cooling and subsurface warming, favouring in turn the melting of basal ice[58,60,61]. Antarctic meltwater supply can also produce stratified surface waters and reduced vertical mixing[24,58,61]. Such enhanced stratification would limit $CO_2$ outgassing to the atmosphere[30] with the latter being indirectly evidenced by reduced carbonate preservation in the ESP record (Fig. 4c). Warming in the SO possibly caused southward shifts of oceanic fronts, the ACC and the SWW. The reduction of ACC strength across the iNHG may have resulted in reduced water exchange across the Drake Passage between the South Pacific and South Atlantic Oceans. The transport of the Antarctic meltwater from the South towards the North Atlantic by the AMOC could have then reduced NADW production[56,59], which would have ultimately weakened the AMOC, reduced the heat supply to the Northern Hemisphere, and favoured the NHG[42]. Although there is currently no proxy evidence for the suggested SO freshening and freshwater forcing on the iNHG due to AIS melting during the long-term SO warming, a recent modelling study on the relation between AIS meltwater and AMOC investigated the role of different locations of meltwater input by hosing experiments in five different sectors of the SO[59]. The authors found that meltwater supply from the West Antarctic marginal seas (i.e. Bellingshausen > Amundsen > Ross Sea), thus close to Site U1543, would have the strongest effect on AMOC weakening.

In addition to a weakened AMOC triggering the iNHG[17,62], decreasing meridional SST gradients in the South and North Pacific caused by high-latitude warming (Fig. 5; ΔSST: U1337–U1543; U1337–ODP882[22]) likely reduced the strength of both the ACC and the PMOC at the Pliocene-Pleistocene transition. A weakened Pacific and SO circulation could have resulted in decreased deep water ventilation, which increased oceanic $CO_2$ uptake[32], thereby contributing to the global cooling during the Pliocene-Pleistocene transition. This would explain the pronounced decrease of reconstructed $pCO_2$ during the iNHG[63] (Fig. 4g). Furthermore, high $pCO_2$ values[22,33,34,63–65] correspond to periods with increased ACC, PMOC, $CaCO_3$ preservation and meridional SST gradients in the South and North Pacific Ocean at least during the Pliocene and early Pleistocene and, thus, could be related to the enhanced global oceanic deep water ventilation (Figs. 4g, 5).

Our study suggests that the pronounced variability in SST, SST gradients and the strength of both the SWW and the ACC in the South Pacific Ocean since the late Miocene was to a substantial part related to shifts in oceanic front's position triggered by AIS fluctuations. During the Pliocene and early Pleistocene, periods with long-term enhanced (decreased) ACC strength were most likely associated with a release (uptake) of $CO_2$ to (from) the atmosphere. Contrasting to the late Pleistocene glacials, when $CO_2$-rich deep water and less ventilation caused carbonate dissolution, during long-term cold periods prior to the late Pleistocene, the generally warmer conditions prevented pronounced sea ice formation and reduced $CO_2$-rich AABW to reach the ESP site. Therefore, in addition to the CSP ACC record[10], the new ESP SST and $CaCO_3$ records, co-varying with ACC from the Pliocene to early Pleistocene, offer additional evidence for the impact of the SO circulation on the global atmospheric $CO_2$ concentration on a 400-kyr

eccentricity timescale[10,36] (Supplementary fig. 2). Changes in ocean circulation are, thus, strongly related to the exchange of carbon between the ocean and atmosphere, not only over glacial-interglacial cycles, but also in a long-term perspective over longer 400-kyr eccentricity cycles. Since the latitudinal extent and position of oceanic fronts are variable in different basins of the SO, the same was possibly true for the ACC along its circumpolar route during the past. Our study also showed a strong 700-kyr lasting warming during the iNHG in the ESP co-varying with other SO SST records. A reduced ACC strength itself that likely benefitted also to SO $CO_2$ uptake, as well as freshwater contribution to the global MOC, was possibly involved in a weakening AMOC that favoured the NHG.

## Methods
### Core lithology and age model
The sediment of the upper 115 m of core U1543 consists of greenish grey to dark greenish grey silt-bearing clay with beds of light grey to light greenish grey carbonate-, clay- and/or diatom-rich nannofossil ooze. Below 115 m, the sediment comprises diatom-bearing silty clay and carbonate as well as silt-bearing diatom ooze[20].

The age model for Site U1543 and PS97 114-2 is based on the magnetic susceptibility tuned to the LR04 $\delta^{18}O$-stack[43] (0–5 Ma) and palaeomagnetic data (5–7.8 Ma) supported by biostratigraphic and palaeomagnetic age control points (Supplementary Fig. 1). The error for biostratigraphic age control points ranges between 0 and 1.03 Ma. The uncertainty of the age model of the LR04 $\delta^{18}O$-stack[43] ranges between 4 kyrs and 40 kyrs[43]. More details of the age model of IODP 383 Sites U1541 and U1543 are available in Lamy et al.[10,20].

### Organic geochemistry
This study presents biomarker analyses of 280 samples of IODP 383 core U1543 taken every ~15 cm covering the period from 8 to 0.5 Ma. From the same site, additional 18 samples of PS97 114-2 core were analysed for the last 0.45 Myrs. According to the current age model[20], this record offers a resolution of 25 kyrs.

Alkenones were extracted from freeze-dried sediments (ca. 4 g) with a mixture of dichloromethane and methanol (DCM/MeOH 9:1, v:v) in an ultrasonic bath. 2-nonadecanone was added as an internal standard. The total lipid extracts were separated into four fractions using hexane, hexane/DCM (1:1), DCM and DCM/MeOH (1:1) using microscale flash column chromatography and silica gel as solid phase[66]. The DCM fraction containing the alkenones was measured with a multichannel TraceUltra GC (Thermo Fisher Scientific) equipped with a VF-200ms column (Agilent Technologies) and a flame ionisation detector. Peak identification was based on comparing peak retention times with an in-house sediment standard. Alkenone-based sea surface temperatures were calculated according to a calibration of the $U^{K'}_{37}$ unsaturation index[67,68].

### Reliability of the $U^{K'}_{37}$ paleothermometer
The long and nearly 8 Myr old record from site IODP383 U1543 might have been affected by changes in alkenone producer assemblages, potentially biasing the $U^{K'}_{37}$ paleothermometer. Therefore, we analysed the relation between the indices of $U^{K'}_{37}$ and $U^{K'}_{38Me}$, which could point to shifts in the algae community[69,70]. The long-term pattern of the $U^{K'}_{37}$ index is very similar to that of the $U^{K'}_{38Me}$ index (Figure. S3b, c) as reflected also by the positive correlation (after removing occasional outliers ($n = 10$), where values are up to twice as high/low as the mean range of 0.1-0.4; Fig. S4), which allows ruling out pronounced shifts in the alkenone producer community as well[69,70].

### Sea surface temperature gradients
The new and other available $SST_{UK'37}$ records[22,23] from the Northern and Southern Hemispheres were used for calculation of SST gradients (Figs. 1, 3, 4). If necessary, SST were recalculated from raw alkenone

data according to the calibration by Müller et al.[68]. All $SST_{UK'37}$ records were 100 kyr-binned by averaging the $SST_{UK'37}$ data for every 100 kyrs in 50 kyr-overlapping windows, allowing calculation and comparison of SST gradients for identical time windows.

## Inorganic geochemistry

The carbonate content ($CaCO_3$) of Site IODP 383 U1543 was determined by linearly calibrating Ca intensities (area counts), obtained through X-ray fluorescence (XRF) scanning with an Avaatech (non-destructive) XRF Core Scanner at IODP TAMU, College Station, USA, with the discrete shipboard $CaCO_3$ measurements performed during IODP Expedition 383[20]. The XRF-scanned Ca intensities were 5-point smoothed and linearly interpolated to the discrete $CaCO_3$ data ($r^2 = 0.923$). The average resolution of the $CaCO_3$ record is 0.33 kyrs. To extend the published 5.3 Ma record of bottom current strength to 8 Ma at IODP Site U1541[10], the calibrated logarithmic count ratio of the XRF-scanned zirconium to rubidium (ln (Zr/Rb)) was used. The detailed methodology for the $ln(Zr/Rb_{XRF})$-based reconstruction is described in Lamy et al.[10].

## Data availability

The data used in the present study are available at the data repository PANGAEA–data publisher for earth & environmental science (https://doi.pangaea.de/10.1594/PANGAEA.982857, https://doi.pangaea.de/10.1594/PANGAEA.982859 and https://doi.pangaea.de/10.1594/PANGAEA.982861).

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

## Acknowledgements

We thank the captain and crew of R/V JOIDES Resolution, as well as scientists and technicians, for their support during IODP Expedition 383 in 2019. Technical support by Jacqueline Hartmann, Nadine Hollmann, and Sascha Plewe is greatly acknowledged. This work was funded by the DFG Priority Programme 527 grants SESPOD (AR 367/16-1; KA 3228/4-1) and IODP383-DYNAPACC (La1273/10-1) to A.W., H.W.A., J.K. and F.L.

## Author contributions

A.W., H.W.A., J.K. and F.L. initiated the project and designed the study. A.W. performed biomarker analyses and drafted the manuscript. F.L. and H.W.A. calculated ACC strength from XRF scanner data and performed the $CaCO_3$ calibration. A.W., H.W.A., J.K., G.W., L.L.-J., V.R., N.R., H.S. and F.L. were closely involved in data interpretation as well as in the preparation and writing of the manuscript.

## Funding

## Competing interests

The authors declare no competing interests.
