## [Transparent Peer Review file · Nature Communications]

South Pacific sea surface temperature and global ocean circulation changes since the late Miocene

Corresponding Author: Dr Antje Wegwerth

Version 0:

Reviewer comments:

Reviewer #1

(Remarks to the Author)

Wegwerth and colleagues present interesting new proxy records from recently drilled IODP cores U1543 and U1541 from the subantarctic Pacific Ocean. Alkenone-derived SST and CaCO₃ contents are presented over the last 8 million years from Site 1543, and an existing Zr/Rb record as proxy for Antarctic Circumpolar Current (ACC) strength is extended for the period 5.3 Ma- 8 Ma at Site U1541. This data is used to discuss on long-term time scales the relation between temperature changes, changes in ACC strength and deep water circulation which influence carbon storage and hence the atmospheric CO₂ content. A focus of the paper is the intensification of the Northern Hemisphere Glaciation (iNHG). During this time the authors describe an unexpected warming, accompanied by a weakening of the ACC and reduced CaCO₃ contents. It is discussed that this favored the iNHG by weakening the AMOC and leading to reduced atmospheric CO₂. While the data presented in this study is an important advance and of interest for a broader readership, the description and comparison of the presented records with other published data should be done more thoroughly. Further, the discussion needs improvement for better explanation, especially the driving mechanisms for a weakening of the AMOC. Figures are in good quality.

Major comments:

The authors state in the abstract and in the last chapter that freshwater pulses into the Southern Ocean and short periods of enhanced ACC weakened the AMOC during the iNHG. However, no clear evidence is provided for this claim. It is known that freshwater input into the North Atlantic reduces the AMOC. In contrast, freshwater input into the Southern Ocean lead there to surface cooling and subsurface warming (Li et al., 2023: cited by the authors). Similar in lines 214-217 it is not clear what is the relationship between water exchange between the South Pacific and Atlantic Oceans and the strength of AMOC. The authors should explain this better.

Periods of long-term warming and cooling are not described correctly by the authors in lines 116-122. For instance, the authors describe a Late Miocene cooling during 8-5.4 Ma, however their SST record in Figure 2b show cooling during ~7.8-6 Ma. The same holds for the described cooling during 5.1-3.7 Ma by the authors. In contrast to this, temperatures are roughly the same at 5.1 and at 3.7 Ma (see Figure 2b). Also, the authors are not consistent in describing the warming period during the iNHG. In line 121 they report a warming during 3.7 Ma-2.3 Ma, whereas in line 132 they describe a warming during 3.2 Ma-2.5 Ma.

The authors describe a cooling at 3.7 Ma as an anomaly as neither observable in the other records in the Southern Hemisphere nor in the global benthic $\delta^{18}\text{O}$ record. However, a cooling of at least 2°C is observed in the NH temperature stack (>50°N) in Herbert et al. (2016) and to a lesser extend in the southern Hemisphere stack (30-50°S).

I think the presented correlation between CaCO₃ contents and AAC strength is convincing and a very important result. However, the correlation between ACC strength and SST is less clear and must be better described. For instance, in lines 144-145 the authors state that on long-term periods ACC strength correlates with their SST reconstructions suggesting stronger ACC when SST cooled. However, during 5-7 Ma I cannot see this correlation between both, and during the late Miocene cooling, the ACC decreased. During 4-3.5 Ma, the ACC increases before the temperature decreases, and also during the last 1.5 Ma, a correlation is not so evident. The same holds for the comparison of the meridional temperature gradients with ACC intensity. Here, the sentence in lines 163-166 is unclear. What mean the authors by "linear relationship"? When there is a good correlation?

More detailed comments:

Line 42: The authors should define the time period of iNHG

Line 70: Delete "regional"

Lines 74-94: Here is repetition and the text can be shortened.

Lines 102-103: What is the role of changes in the primary productivity for carbonate preservation? (Rea and Leinen, 1985).

Lines 106-107: Maybe it would be more straightforward if the authors mention other studies which already used this proxy before.

Lines 110-112: Isn't it also sensitive to changes in the fronts?

Line 114: Please indicate Fig. 2b

Line 218: When the iNHG was "preconditioned" by increased AMOC? What does this mean?

Methods:

Please define the error in the used age model.

Which calibration is used for the calculation of SST?

Line 284: The authors should mention the resolution of the CaCO₃ measurements, and the error should be provided.

Figures:

Figure 2b: It should be mentioned how the smooth was calculated.

Figure 3b: The same as above.

Reviewer #2

(Remarks to the Author)

South Pacific sea surface temperature and ocean circulation changes since the Late Miocene

Wegwerth and coauthors reconstructed the sea surface temperature (SST), the strength of the Antarctic Circumpolar Current (ACC), and carbonate preservation at two sites from the central and eastern South Pacific over the past eight million years. The SST records reveal a long-term cooling trend from the Miocene to the Pleistocene. Notably, there is pronounced cooling at approximately 3.7 Ma and warming from 3.2 Ma to 2.5 Ma, both of which strongly differs from those as indicated by the global benthic oxygen records. The authors show by combining SST records from the eastern equatorial Pacific increased meridional SST gradient during cooler periods, which favours strengthened ACC driven by intensified westerly winds, and vice versa. Going along with the variations in the ACC, deep waters release increased (decreased) CO₂ to the atmosphere during cooling (warming) periods.

This manuscript presents various proxies for the SST variability, the ACC strength and oceanic carbonate preservation, which advances our understanding of the variations in the atmospheric and oceanic circulation, and their interactions with the global carbon cycle in the past. Yet there are some concerns that should be addressed so I recommend a major revision.

Major Comments

1) Lines 178 to 195

In this paragraph, the authors argue based on proxies and model simulations that the strengthened ACC may have served as a potential driver for an active Pacific Meridional Overturning Circulation (PMOC) over the last 5.9 million years. Yet I feel that the referenced model simulations do not support this argument.

Indeed, model simulations by Burls et al. (2017) are able to show deep water formation over the subarctic North Pacific, and thus a deep PMOC extending down to ~3000 m under Pliocene boundary conditions. However, such a deep PMOC is obtained by imposing large perturbations in the top-of-the-atmosphere radiation budget via substantial modifications in the cloud albedo (Burls et al. 2017). The activation of PMOC should therefore be viewed as a response to the imposed radiative forcing, yet not to changes in the ACC. Global warming simulations performed Burls et al. 2017 support that radiative forcing is crucial to an active and deep PMOC. The other reference by Fu et al. (2024) does not support the influence of ACC on the PMOC, either. To obtain an active PMOC, Fu et al., (2024) imposed strong freshwater forcing amounting to 0.4 Sv over the subarctic North Pacific, which may lead to the strengthened ACC. The referenced model simulations both support that a strongly reduced halocline over the subarctic North Pacific is a prerequisite for the deep water formation and hence a deep PMOC. However, I feel that neither of them support the influence of ACC on the PMOC.

Moreover, we should note that the PMOC during the last deglaciation extends down ~ 2000 m, which is approximately 1000 m shallower than that during the Pliocene. To me, this should be referred as the ventilation of intermediate waters, yet not deep water ventilation.

2) Lines 186-188

"... thereby causing a stronger bifurcating of the ACC in vicinity of the South American continent with an intensified northward directed Humboldt Current system as an integral component of the Pacific Ocean Circulation."

I understand that the strengthened ACC could enhance the northward Humboldt Current. Yet please explain how it is linked to PMOC.

3) Lines 227-229

I found the statement "A further study on the impact of Antarctic and Greenland meltwater on future climate suggests that Antarctic meltwater will have a dominant effect on atmospheric cooling and the AIS" confusing.

As I understand, the study by Li et al. (2023) suggests a dominant impact of Antarctic meltwater on the atmospheric cooling in the Southern Hemisphere. The meridional overturning circulation and atmospheric cooling in the Northern Hemisphere is dominated by Greenland meltwater.

4) Lines 230-233

“... decreasing meridional SST gradients in the South and North Pacific caused by high-latitude warming (Supplementary Fig. 3; Δ SST: U1337 – U1543; U1337 – ODP88213) likely reduced the strength of both the ACC and the PMOC.”

I feel the relation between the PMOC strength and meridional SST gradient in the North Pacific is unclear. For instance, when meridional SST gradient in the North Pacific reached its peak at around 2.7-2.8 Ma, the PMOC strength as indicated by carbonate preservation reached its minimum (Figure S3d and S3f). Moreover, the variations in the PMOC strength hardly exhibit relations to the meridional SST gradient in the North Pacific from 4 Ma to 3 Ma.

Minor Comments:

1) Lines 67, 70, 99 and others

The authors use several expressions for million years, such as Myrs and Ma. Please consider using only one abbreviation.

2) Line 141

The word “Circulation” seems to be redundant.

3) Lines 161 and 169 “central equatorial Pacific”

I feel it is better to use “eastern equatorial Pacific”, as U1337 is located in the realm of the equatorial Pacific cold tongue.

4) Line 413

It should be reference # 36.

5) Line 505 Caption for Figure 3

Lines 540 and 541 Caption for Figure S3

Line 553 Caption for Figure S4

It should be “eastern equatorial Pacific”.

Reviewer #3

(Remarks to the Author)

The paper presents a crucial new record of sea surface temperature in the southeast Pacific over the late Neogene. Combined with previously published grain size data for the Neogene from another site (Lamy et al., 2024), it documents the local ocean temperature change in a crucial sector of the Southern Ocean, where the ACC interacts with geographic obstructions. It documents peculiar gradual shifts in regional SST trends that are decoupled from those in the benthic foraminifer isotope trends. This in itself is a really interesting result, and I congratulate the authors for putting this together. By comparing this new record to some other SST records in the Southern Ocean (although several crucial datasets are unfortunately omitted), delivers a reconstruction of South Pacific temperature and ocean circulation changes, that, according to the authors is fully centered around variations in strength of the ACC. Although the record of sortable silt does express convincingly dramatic changes in bottom ocean flow at this site, I am not convinced of the somewhat unidirectional interpretation that that reflects only strength in ACC flow.

I find the paleotemperature record fascinating, but I have some technical questions regarding the smoothing of the data that I need to be informed of. I will elaborate my concerns below, but in short, the manuscript should:

-put their results in the context of recent insight into the development of the ACC (Evangelinos et al., 2023)

-be complete in the integration of existing SST datasets that cover this time and are from the Southern Ocean, to complete the picture of SST developments

-explain how their interpretation of ocean circulation changes and ACC strength fits into observations of a DECREASING meridional SST gradient from the subtropics to Antarctica throughout the Neogene (Hou et al., 2023a)

-give proper attention to the fact that not the ACC but ocean fronts reflect meridional SST gradients, and that these shifted in latitude profoundly through geologic time (Hou et al., 2023b). The authors must convincingly show that their reconstructions are indeed a sign of ACC strength rather than latitudinal shifts, and properly explain why their records argue against latitudinal shifts in fronts, or properly explain in the paper why they believe these frontal shifts are not important in explaining the trends they see. Finally they must explain how variations in ACC strength can induce such strong local SST variability as they see in their record.

Context of other papers on meridional temperature gradient development

Recent efforts have shown the complexity of the developments in the Southern Ocean oceanography, with dating the onset of modern-like ACC flow younger and younger (Evangelinos et al., 2023), in spite of absence of geographic barriers, and with development of the latitudinal temperature gradient in the Southern Ocean contrasting to what one would expect (Hou et al., 2023a). Also previous work has demonstrated the strong mid-latitude cooling in the Southern Ocean, and the effect of polar amplification in the development of the Neogene subtropical gyre (Liu et al., 2022). In light of these papers, none of which are cited in this paper, I am unsure what exactly the innovation of this paper is, although new SST records showing regional SST trends, are always insightful. For instance, the authors mention in the introduction that the westerly winds and ACC strength are strongly governed by the latitudinal temperature gradient. In the discussion I understand that this refers to the meridional temperature gradient of the subtropical gyre, and not that of the Southern Ocean itself. Still, Hou et al., 2023a showed in the compilation of SST records a decrease in latitudinal temperature gradient between the subtropical front and the Antarctic continental margin since the middle Miocene. How do the authors reconcile this with their statements? Is the meridional temperature gradient not set by the position of the fronts which, in turn are governed by the strength and position of the westerly winds? How is the ACC affected by the fact that the subpolar meridional SST gradient, the one where the

ACC is actually in, decreases over the Neogene? Perhaps there is a logical answer to this, but I find this aspect missing in the paper altogether.

Incomplete comparison to other records

It is strange that the high-resolution SST data from Site 594, 806, (Herbert et al., 2016) The record from the Ross Sea (Duncan et al., 2023) and Site 1168 (Hou et al., 2023a Climate of the past) are missing in this paper. Besides those papers paving the way for the inferences made in this paper, they provide crucial clues about the development of meridional temperature gradients and frontal systems (particularly in Hou et al., 2023b nature communications). Even if there are concerns to compare SST records from different proxies, Site 1168 has a convincing comparison between UK37 and TEX86-based SST and thus TEX86 can be used as SST proxy without doubt. Site 594 and those of the western Pacific warm pool are from UK37.

ACC strength versus latitudinal shifts

My other main concern is that this paper unidirectionally points to ACC strength variability as a means to explain the data, and completely seems to ignore changes in the latitudinal position of the ACC and the fronts and the local oceanographic effect of those. There is an overwhelming amount of evidence from the modern (e.g., a review in Chapman et al., 2020), Pleistocene (Bard and Rickaby, 2009; Ai et al., 2023) and Neogene (Hou et al., 2023b) that fronts, which determine the meridional temperature gradients much more than the ACC, shift in position, due to changes in the position of the westerlies (Graham et al., 2012). There is another reason to assume more influence of frontal system shifts driving the signals in the technical notes on the SST curve below.

Technical aspects of the new SST curve

The temporal resolution of the new data is impressively high but the SST data itself is also quite variable from sample to sample, with amplitudes of variability of up to 5 degrees. First of all, I note that the frequency of that variability is not fully captured by the temporal resolution of the data, and this means that the amplitude of variability might be underestimated, which may affect adequate presentation of the trends. Secondly, and this relates to my point made previously, this variability demonstrates that the majority of the oceanographic variability at this site reflects a latitudinal shift in fronts. SST decreases rapidly per latitude across these fronts. As a result, local SST variability is profoundly amplified as the front shifts over the site. Climate forcing alone cannot explain such high variability in the SST. In light of this I do not understand why all emphasis in this paper is put on variability in strength. I am not denying that there was any change in strength of the ACC, I am just surprised how little focus there is in the paper on this factor. If the ACC strength was the only thing that varied, why would SST be so variable? The zonal flow of the ACC would not be able to change local SST just by changing in strength, right? The authors must acknowledge that the amplitude of variability in their SST record reflects ocean frontal shifts, and note that the temporal resolution is unable to fully capture the cyclicity which causes underestimation of the amplitude of variability AND may have some influence on the expression of the trends.

Minor points:

Abstract: Abstract does not specify what data is really new in terms of data, or reconstruction

Main text: "Late Miocene" is not a formal time zone in the Geologic time scale, and thus "Late" should not be capitalized
Lines 102/103: I find the explanation of the CaCO₃ records and what they mean in terms of oceanography a bit too simple. For instance: it doesn't give credit to the fact that surface ocean carbonate production changes strongly in the different ocean zones. The carbonate factory may, in a deep ocean with corrosive bottom waters, play a large role in determining the amount of calcium carbonate preservation. The text and referencing at this part is quite general and not directly geared towards the local processes at hand. I am not convinced that the local calcium carbonate record indeed reflects the amount of global ocean CO₂ outgassing as the authors suggest. At least, the authors have not fully excluded alternative hypothesis that may have driven calcium carbonate preservation at this site.

Line 113: can community shifts in alkenone producers be excluded? Is there a record of the C38me alkenones to confirm this?

Line 136: how does this play out in Rohling et al's more recent, and more sophisticated deconvolutions of the benthic isotope stack?

Line 220: very speculative. First of all, the Antarctic peninsula never had much ice to melt, which limits the amount of freshwater delivery. Second of all it is unsure how far that meltwater travelled, third of all, it may only locally have reduced CO₂ outgassing, with unknown feedbacks in the rest of the ocean. Plus if there is one "machine that makes the freshening effect of meltwater disappear it is the deep-reaching ACC.

Line 113–122: Main point that demonstrates the importance of the position of ocean zones and fronts: if one was to translate these coolings, in comparison to the atmospheric CO₂ decline, to climate sensitivity and polar amplification (somewhere between 2 and 4 per CO₂ doubling and somewhere between 2 and 3 x times global average), one would arrive at ridiculously high numbers none of which are confirmed by model or climate theory. Conclusion: local ocean conditions must have changed such that ocean temperature change was amplified compared to the climate forcing. In a latitudinal belt of the strongest latitudinal temperature gradient, the easiest way to amplify climate change is by latitudinal shifts of ocean zones.

Line 226: Note that that freshening effect came about by a episodic release from a HUGE ice sheet, in a MUCH smaller ocean basin without an ACC. The analogy to the southern hemisphere is not so clear to me.

Ai, X.E., Thöle, L.M., Auderset, A. et al. The southward migration of the Antarctic Circumpolar Current enhanced oceanic degassing of carbon dioxide during the last two deglaciations. *Commun Earth Environ* 5, 58 (2024).

<https://doi.org/10.1038/s43247-024-01216-x>

Bard, E., Rickaby, R. Migration of the subtropical front as a modulator of glacial climate. *Nature* 460, 380–383 (2009).

<https://doi.org/10.1038/nature08189>

Duncan, B., McKay, R., Levy, R. et al. Climatic and tectonic drivers of late Oligocene Antarctic ice volume. *Nat. Geosci.* 15, 819–825 (2022). <https://doi.org/10.1038/s41561-022-01025-x>

Evangelinos, D., Etourneau, J., van de Flierdt, T. et al. Late Miocene onset of the modern Antarctic Circumpolar Current. *Nat. Geosci.* 17, 165–170 (2024). <https://doi.org/10.1038/s41561-023-01356-3>

Graham, R. M., A. M. deBoer, K. J. Heywood, M. R. Chapman, and D. P. Stevens (2012), Southern Ocean fronts: Controlled by wind or topography?, *J. Geophys. Res.*, 117, C08018, doi:10.1029/2012JC007887.

Herbert, T., Lawrence, K., Tzanova, A. et al. Late Miocene global cooling and the rise of modern ecosystems. *Nature Geosci* 9, 843–847 (2016). <https://doi.org/10.1038/ngeo2813>

Hou, S., Lamprou, F., Hoem, F. S., Hadju, M. R. N., Sangiorgi, F., Peterse, F., and Bijl, P. K.: Lipid-biomarker-based sea surface temperature record offshore Tasmania over the last 23 million years, *Clim. Past*, 19, 787–802, <https://doi.org/10.5194/cp-19-787-2023>, 2023a

Hou, S., Stap, L.B., Paul, R. et al. Reconciling Southern Ocean fronts equatorward migration with minor Antarctic ice volume change during Miocene cooling. *Nat Commun* 14, 7230 (2023b). <https://doi.org/10.1038/s41467-023-43106-4>

Liu, X., Huber, M., Foster, G.L. et al. Persistent high latitude amplification of the Pacific Ocean over the past 10 million years. *Nat Commun* 13, 7310 (2022). <https://doi.org/10.1038/s41467-022-35011-z>

Peter Bijl

Version 1:

Reviewer comments:

Reviewer #1

(Remarks to the Author)

Wegwerth et al. sufficiently addressed many of my former comments in the revised version. However, before possible publication, the authors need to clarify following remaining issues:

The authors now explain much better the possible effects of a freshening in the Southern Ocean on the Atlantic Meridional Overturning Circulation (AMOC) during the iNHG. However, as the authors state that there is yet no proxy evidence for this hypothesized freshening, the whole section dealing with this topic (lines 357-374) is speculative and should be shortened. If the authors could show a proxy record demonstrating significant AIS melting within a region suggested to be critical in affecting the AMOC, this claim would be much stronger.

The proposed Southern Ocean warming during the iNHG is better elaborated in the revised version. Wegwerth and colleagues now show more SST proxy records from this region. However, the discussion of these SST proxy records still needs to be improved. For example, in lines 324-326 the authors state that SST proxy records from the Southwest Pacific and from the South Atlantic show a warming of about 3 °C during the iNHG. However, the SST records from the Southwest Pacific seen in Figure 3f and g show stable SST or a cooling during 3-2.3 Ma.

The presented correlation of CaCO₃ preservation in the deep North Pacific and the ACC is very interesting and worth mentioning. However, the discussed influence of the ACC through the Humboldt Current towards the deep North Pacific in lines 302-306 is rather unclear. The model simulation, mentioned in line 51 which is presented to support this link explains that primarily increased salinities in the North Pacific caused an increased Pliocene PMOC and that this led to a strengthened Drake Passage throughflow. I suggest, the authors better describe the connection between the ACC and the North Pacific through the PMOC.

Detailed comments:

Lines 63-65: I suggest mentioning the Southern Westerly Winds first.

Line 206: “reduced SST” within the ACC?

Line 350 and following: What is the effect of meltwater on reconstructed alkenone-derived SST?

Reviewer #2

(Remarks to the Author)

Overall Assessment

The authors have addressed my comments raised in the first round of review and revised the manuscript accordingly. The

long-term interplay among the sea surface temperature, CaCO₃ preservation, the ACC as indicated by proxies in the Southern Ocean and Southern Hemisphere Westerlies are well explained.

There are a few minor points that I would like to stress upon (the line numbers are from the revised manuscript with track-changes).

Comments

Lines 75-77

I find this statement confusing. The final closure the Panama Seaway around 5 Ma is indeed responsible for the intensification of the AMOC, yet it weakens the ACC as shown by Figure 7d in Huang et al., (2024). The authors may consider revising this sentence.

Lines 170

To avoid confusion, would it be better to move the reference # 36 after the word "record"?

Lines 175

The figure citation is incorrect. It should be Fig. 2c, yet not Fig. 1c.

Line 179

As shown in Fig. 2d, the SSTs increased by approximately 6°C from 3.0 Ma to 2.3 Ma. This sentence may be expressed as something like "SSTUK'37 rose to 12°C reaching ..."

Line 358

To avoid confusion, I prefer replacing "fresher water" with "the Antarctic meltwater?"

Lines 369-374

I could not agree that the high quantity release of the AIS freshwater to the SO exceeded the modelled threshold. As stated in An et al., (2024), a meltwater input of 1.0 Sv (1 Sverdrup is equivalent to 10⁶ m³/s) into the ocean over 100 years accounts for about 10% of the current total Antarctic ice sheet. If the meltwater from the AIS reached the volume threshold, i.e., 0.1 Sv, then the entire AIS would melt after approximately 10,000 years. However, the authors investigate a long-term warming of about 700 kyrs. The quantity of meltwater must be much smaller than the volume threshold.

Reviewer #3

(Remarks to the Author)

I have re-reviewed this manuscript after a round of profound revisions. My comments and that of other reviewers have been carefully and dilligently incorporated and I have no further comments to make. The manuscript now reads much better and puts the study in a more complete context in the region.

Version 2:

Reviewer comments:

Reviewer #1

(Remarks to the Author)

The authors have considered all my remaining comments, and I do not have more observations.

Reviewer #2

(Remarks to the Author)

The authors have well addressed most of my concerns. This study is a very good contribution to advance our understanding about the past changes in the ACC and Southern Ocean dynamics.

I still have four some comments for the authors to consider. Yet it appears unnecessary to review the revised manuscript for an additional round.

The line numbers are from the revised manuscript.

1. Lines 64 – 67

"The intensity of the ACC depends on the strength of the Southern Westerly Winds (SWW) and the large-scale density structure of the SO, influenced by the extent of the Antarctic Ice Sheet (AIS) and sea ice. Both influence the position of the frontal system and are controlled by changes in equator-to-pole temperature gradients."

If I understand correctly, the word "both" here refer to the Southern Ocean westerly and the density structure of the SO, but not the AIS and sea ice. The authors may consider revise the words to avoid confusion.

2. Line 96

"... triggers a strengthening ocean circulation and rise of atmospheric CO₂, which in turn results in warming ..."

Would it be better to replace "results in" with "further amplifies"?

3. Lines 97-99

"... the interplay between climate, ocean circulation, and CO₂ changes, ..."

Would it be better to replace "between" with "among"?

4. Lines 306-307

“Therefore, an intensified PMOC might not have only favoured an enhanced Drake Passage throughflow, but also a strengthened ACC at least in the Pacific.”

I would change the text to “an intensified PMOC might not only go along an enhanced Drake, ...”.

The intensified PMOC and strengthened ACC are both linked to the freshwater forcing in the simulations. It is difficult to conclude that the enhanced PMOC favours a stronger ACC.

5. Data availability

It is better to add a link to the Pangaea repository.

Reviewer #3

(Remarks to the Author)

I re-reviewed the manuscript, and although I did not have much to comment on in the revisions last time, I do have two minor comments.

In the introduction the authors make some strong connections between ACC strength, Panama isthmus closure and AMOC developments, and although the cited papers do provide some evidence for these connections, I think that the relationships and causalities between these processes are not yet so firmly established. The authors must be careful in stating these causalities, to reflect the ongoing research on these processes.

I also have a concern with the new presentation of the SST trends derived from Fig. 3 around the iNHG. I do not see the consistency between the records that the authors propose. The interpretation of warming in other records than the new record of the authors (although also THAT warming is not unique in their record) is not clear. There may be sustained warmth in that time interval in other records, though not unique for that time interval. The apparent assumption that the authors make is that the Southern Ocean should cool or warm uniformly (they see that the warming occurs consistently in all SST records), but it is clearly not uniform and I do not understand why the authors would think that it should be. The Southern Ocean does not have a lot of latitudinal exchange anymore by that time, and so different ocean zones can behave differently, as a result of changes in local conditions or atmospheric heat redistribution. The authors should reflect that better in the inserted part, because I do not necessarily disagree with their review of studies that the AA ice sheet was reduced during the iNHG, or that frontal systems were displaced southwards (Hou et al., 2023 Nature Communications shows that the southward shift of the STF started already by 4 Ma), I am just saying that the manuscript now seems to review the available SST data in search of a uniform warming, and I think that is not required to make that case.

REVISION NOTES: Ref. No.: NCOMMS-24-56160-T

The revised parts in the text are here marked in **bold**. Longer revised paragraphs are referred by **line numbers** only.

→ Changes in the tracked changes version are highlighted in **yellow**

Reviewer #1 (Remarks to the Author):

“Wegwerth and colleagues present interesting new proxy records from recently drilled IODP cores U1543 and U1541 from the subantarctic Pacific Ocean. Alkenone-derived SST and CaCO₃ contents are presented over the last 8 million years from Site 1543, and an existing Zr/Rb record as proxy for Antarctic Circumpolar Current (ACC) strength is extended for the period 5.3 Ma- 8 Ma at Site U1541. This data is used to discuss on long-term time scales the relation between temperature changes, changes in ACC strength and deep water circulation which influence carbon storage and hence the atmospheric CO₂ content. A focus of the paper is the intensification of the Northern Hemisphere Glaciation (iNHG). During this time the authors describe an unexpected warming, accompanied by a weakening of the ACC and reduced CaCO₃ contents. It is discussed that this favored the iNHG by weakening the AMOC and leading to reduced atmospheric CO₂. While the data presented in this study is an important advance and of interest for a broader readership, the description and comparison of the presented records with other published data should be done more thoroughly. Further, the discussion needs improvement for better explanation, especially the driving mechanisms for a weakening of the AMOC. Figures are in good quality.”

We thank the Reviewer#1 for the positive assessment of the manuscript and appreciation of the new dataset. We are grateful for the valuable suggestions regarding a better explanation of our data and inferred mechanisms as well as for pointing us to a more detailed comparison with other published data. We have followed these suggestions as shown in the point-by-point response to the Reviewer #1's comments. In short, we added a stacked global and Southern Hemisphere temperature record (Clark et al., 2024) to Fig. 2 as well as five SST records from the Southern Ocean and the equatorial Pacific to original Fig. S2 (now main Fig. 3), which are now involved in the discussion. Regarding the AMOC weakening potentially favoured by warming in the Southern Ocean during the iNHG, we extended the discussion by additional evidence for warming in the Southern Ocean and a reduced Antarctic Ice Sheet during that time window. While the reduced ACC strength by itself likely weakened the AMOC, we propose that enhanced surface freshwater transport to the North Atlantic could have caused reduced North Atlantic deep-water formation thereby contributing to a weakened AMOC. We further cited a recent study by An et al. (2024), which supports AMOC weakening by Antarctic meltwater supply.

“Major comments:

The authors state in the abstract and in the last chapter that freshwater pulses into the Southern Ocean and short periods of enhanced ACC weakened the AMOC during the iNHG. However, no clear evidence is provided for this claim. It is known that freshwater input into the North Atlantic reduces the AMOC. In contrast, freshwater input into the Southern Ocean

lead there to surface cooling and subsurface warming (Li et al., 2023: cited by the authors). Similar in lines 214-217 it is not clear what is the relationship between water exchange between the South Pacific and Atlantic Oceans and the strength of AMOC. The authors should explain this better.”

We thank the reviewer for pointing us to better explain the relation between Antarctic meltwater input into the Southern Ocean and AMOC intensity. We have revised the entire chapter by adding more information on our proposed mechanisms accounting for the suggested interplay between Antarctic ice-sheet (AIS) melting and Atlantic Meridional Overturning (AMOC) weakening. We also reorganised the respective chapter in order to improve the wording and succession of our arguments and by discussing additional published studies on AIS meltwater forcing on AMOC weakening.

We removed the argument that intermittent enhanced ACC strength weakened the AMOC. Instead, the reduced ACC strength itself connecting the Pacific and Atlantic Ocean might have resulted in a general slow-down of the global MOC.

Besides the evidence for a highly dynamic AIS during the iNHG, which was likely related to episodic meltwater release, we now also show other SST records (new main figure 3) in the Southern Ocean (i.e. South Atlantic, Western South Pacific) that clearly show warming during the iNHG, which would suggest general AIS meltwater release also beyond our study area. Therefore, we expect increased transport of freshwater towards the North Atlantic that consequently reduced NADW production and hence hampered AMOC (Stouffer et al., 2007; An et al., 2024). The resulting limited heat supply to the northern hemisphere could have favoured the build-up of northern hemisphere ice sheets. Finally, we mention a recent modelling study, which demonstrates that meltwater supply, particularly from the ESP, has a pronounced effect on AMOC weakening (An et al., 2024). Despite these more comprehensive data sets, direct proxy evidences for freshening would be needed to further support our theory, but these are not available.

“Periods of long-term warming and cooling are not described correctly by the authors in lines 116-122.”

We thank the reviewer for pointing out our inconsistent description of the individual time frames with warming and cooling, which resulted from the intention to provide a summarised overview in a long-term perspective. We carefully went through the entire manuscript and now consistently named the respective time intervals and SST amplitudes accordingly.

“For instance, the authors describe a Late Miocene cooling during 8-5.4 Ma, however their SST record in Figure 2b show cooling during ~7.8- 6 Ma. The same holds for the described cooling during 5.1-3.7 Ma by the authors. In contrast to this, temperatures are roughly the same at 5.1 and at 3.7 Ma (see Figure 2b). Also, the authors are not consistent in describing the warming period during the iNHG. In line 121 they report a warming during 3.7 Ma-2.3 Ma, whereas in line 132 they describe a warming during 3.2 Ma-2.5 Ma.”

We corrected the inconsistent time periods of long-term SST changes in lines 157-163.

According to the revised periods, we corrected the arrows in our new Figure 2d.

“The authors describe a cooling at 3.7 Ma as an anomaly as neither observable in the other records in the Southern Hemisphere nor in the global benthic $\delta 18O$ record. However, a cooling of at least 2°C is observed in the NH temperature stack (>50°N) in Herbert et al. (2016) and to a lesser extent in the southern Hemisphere stack (30-50°S).”

It is correct that the 3.7 Ma cooling is also seen in other regional records, but not as prominent as in our ESP record. However, we revised the paragraph in lines 171-175, and now acknowledge that other SST records from both hemispheres imply only less cooling and added more supporting references. These records are shown in the revised Figure S2 (now main figure 3).

“I think the presented correlation between CaCO₃ contents and ACC strength is convincing and a very important result. However, the correlation between ACC strength and SST is less clear and must be better described. For instance, in lines 144-145 the authors state that on long-term periods ACC strength correlates with their SST reconstructions suggesting stronger ACC when SST cooled. However, during 5-7 Ma I cannot see this correlation between both, and during the late Miocene cooling, the ACC decreased. During 4-3.5 Ma, the ACC increases before the temperature decreases, and also during the last 1.5 Ma, a correlation is not so evident. The same holds for the comparison of the meridional temperature gradients with ACC intensity. Here, the sentence in lines 163-166 is unclear. What mean the authors by “linear relationship”? When there is a good correlation?”

We agree that the sign of correlations between ACC and SST is not consistent on different time-scales and during different periods. Therefore, we originally mentioned that the ACC increasing strength during cooling intervals is mostly seen for the Pliocene and early Pleistocene period (iNHG), now incorporated throughout the manuscript. Furthermore, we explicitly referred to the long-term trend of the SST record and not to individual interglacials and glacials because of the too low resolution for a comparison on orbital or multi-millennial timescales. Nevertheless, we went through the respective paragraph in lines 188-192, described the above-mentioned pattern more clearly, revised our original statements, indicated exceptions, and reworded relevant parts accordingly.

Additionally, we revised the SST panel in Figure 3a (now Fig. 4a), since we realised that the smoothed (i.e., 100 kyr binned) SST record is less correlated with the ACC strength than the unsmoothed SST dataset. Therefore, we removed the binned SST record from that figure. As we explicitly focus on long-term trends, we revised the arrangement of the grey bars in the figure, which represent periods of long-term intensification of the ACC. We further added 100 kyr binned records of ACC strength and CaCO₃, which supports the correlation to the SST gradient reconstruction and shows increased values during almost each period (grey bars) of high ACC strength (exceptions are only seen during the late Miocene at 6.2 Ma and 6.7 Ma). We removed “linear relation” from the respective sentence, but point to the similar patterns of the proxy records.

“More detailed comments:

Line 42: The authors should define the time period of iNHG“

We define the iNHG not in the abstract, but in the main text, where it is first mentioned in lines 178.

“Line 70: Delete “regional” “

Corrected.

“Lines 74-94: Here is repetition and the text can be shortened. “

We reorganised the paragraph to avoid repetition. Since we added some more important information to the introduction chapter, this paragraph may appear as long as in the original version (lines 105-115).

“Lines 102-103: What is the role of changes in the primary productivity for carbonate preservation? (Rea and Leinen, 1985). “

We agree with Reviewer #1 that the carbonate contents may reflect both productivity and preservation, but that preservation is the main factor at Site U1543 (Kasuya et al., 2024). This is now included in the main text (lines 133-134, 140-143). We added the proposed reference.

“Lines 106-107: Maybe it would be more straightforward if the authors mention other studies which already used this proxy before. “

We agree and added the respective references. Additionally, we explicitly mention a record of last glacial/interglacial cycles at Site U1543 (lines 140-143).

“Lines 110-112: Isn't it also sensitive to changes in the fronts“

We added more information on fronts now in the introduction in lines 65-72, 77-82, and 104. In addition, we discuss more thoroughly the role of frontal movements in SST and ACC changes (205-229, 271, 337, 355). Furthermore, we added the modern configuration of Southern Ocean Fronts to the main Fig. 1.

“Line 114: Please indicate Fig. 2b“

It has been added (now Fig. 2d).

“Line 218: When the iNHG was “preconditioned” by increased AMOC? What does this mean“

According to Haug and Tiedemann (1998), an intensified AMOC transported enhanced moisture/rainfall to the Northern Hemisphere, which favoured (preconditioned) the built-up of the Northern Hemisphere Ice Sheet. Due to the revision of the entire paragraph, the sentence is now removed.

“Methods:

Please define the error in the used age model. “

The age uncertainties of the LR04 $\delta^{18}\text{O}$ -stack have been quantified to range from 4 to 40 kyrs. Our graphical correlation of the magnetic susceptibility record to the LR04 $\delta^{18}\text{O}$ -stack is difficult to quantify, but lies probably in the same range. The age uncertainties of

the paleomagnetic and biostratigraphic age control points are partly larger, but are only used to confirm the plausibility of the graphical tuning (lines 414-419).

“Which calibration is used for the calculation of SST”

We used the calibration from Müller et al. (1998), which is cited in the original manuscript by reference no. 50 and now reference 70.

“Line 284: The authors should mention the resolution of the CaCO₃ measurements, and the error should be provided.”

We added the respective information and now it reads in line 463: **“The average resolution of the CaCO₃ record is 0.33 kyrs.”** We also added the correlation coefficient ($r^2=0.923$) between Ca intensities based on XRF and discrete CaCO₃ measurements, which supports the methodology.

“Figures:

Figure 2b: It should be mentioned how the smooth was calculated.”

This has been added to the text (lines 772-774).

“Figure 3b: The same as above.”

As mentioned above, we removed the smoothed SST graph from that figure allowing a better comparison with the ACC strength.

Reviewer #2 (Remarks to the Author):

“South Pacific sea surface temperature and ocean circulation changes since the Late Miocene

Wegwerth and coauthors reconstructed the sea surface temperature (SST), the strength of the Antarctic Circumpolar Current (ACC), and carbonate preservation at two sites from the central and eastern South Pacific over the past eight million years. The SST records reveal a long-term cooling trend from the Miocene to the Pleistocene. Notably, there is pronounced cooling at approximately 3.7 Ma and warming from 3.2 Ma to 2.5 Ma, both of which strongly differs from those as indicated by the global benthic oxygen records. The authors show by combining SST records from the eastern equatorial Pacific increased meridional SST gradient during cooler periods, which favours strengthened ACC driven by intensified westerly winds, and vice versa. Going along with the variations in the ACC, deep waters release increased (decreased) CO₂ to the atmosphere during cooling (warming) periods.

This manuscript presents various proxies for the SST variability, the ACC strength and oceanic carbonate preservation, which advances our understanding of the variations in the atmospheric and oceanic circulation, and their interactions with the global carbon cycle in the past. Yet there are some concerns that should be addressed so I recommend a major revision.”

We appreciate the positive feedback by Reviewer#2 regarding the value of the presented proxies for understanding the ocean-atmosphere system and the carbon cycle back to the late Miocene.

“Major Comments

1) Lines 178 to 195

In this paragraph, the authors argue based on proxies and model simulations that the strengthened ACC may have served as a potential driver for an active Pacific Meridional Overturning Circulation (PMOC) over the last 5.9 million years. Yet I feel that the referenced model simulations do not support this argument.

Indeed, model simulations by Burls et al. (2017) are able to show deep water formation over the subarctic North Pacific, and thus a deep PMOC extending down to ~3000 m under Pliocene boundary conditions. However, such a deep PMOC is obtained by imposing large perturbations in the top-of-the-atmosphere radiation budget via substantial modifications in the cloud albedo (Burls et al. 2017). The activation of PMOC should therefore be viewed as a response to the imposed radiative forcing, yet not to changes in the ACC. Global warming simulations performed Burls et al. 2017 support that radiative forcing is crucial to an active and deep PMOC. The other reference by Fu et al. (2024) does not support the influence of ACC on the PMOC, either. To obtain an active PMOC, Fu et al., (2024) imposed strong freshwater forcing amounting to 0.4 Sv over the subarctic North Pacific, which may lead to the strengthened ACC. The referenced model simulations both support that a strongly reduced halocline over the subarctic North Pacific is a prerequisite for the deep water formation and hence a deep PMOC. However, I feel that neither of them support the influence of ACC on the PMOC.

Moreover, we should note that the PMOC during the last deglaciation extends down ~ 2000 m, which is approximately 1000 m shallower than that during the Pliocene. To me, this should be referred as the ventilation of intermediate waters, yet not deep water ventilation. “

We thank the reviewer for this relevant comment. In order to revise our discussion on the relation between PMOC and ACC, we rewrote the respective paragraph (lines 289-317) and also removed more speculative statements such as the ACC as a potential driver of the PMOC. We, nevertheless, see some noteworthy similarities in the patterns of SST changes, and especially of the CaCO₃ preservation records in the North and South Pacific sites (now better seen by the 100-kyr binned proxy records in Figure S3; now new main Fig. 5) with the latter suggesting enhanced deeper ocean circulation and ventilation. Therefore, though not proven in detail by proxy records or model simulations, we would like to point to the possibility of a mechanistic relation between PMOC and ACC. We further refer to a recent study, which points to the dependence of an enhanced AMOC on ACC (Huang et al., 2024), in agreement with our concept.

“2) Lines 186-188

“... thereby causing a stronger bifurcating of the ACC in vicinity of the South American continent with an intensified northward directed Humboldt Current system as an integral component of the Pacific Ocean Circulation.”

I understand that the strengthened ACC could enhance the northward Humboldt Current. Yet please explain how it is linked to PMOC.”

We added the sentence and now it reads in lines 305-306: **“The latter flows towards the equatorial Pacific and comes upon the westward equatorial current, thus linking the Southern with the Northern Pacific.”**

“3) Lines 227-229

I found the statement “A further study on the impact of Antarctic and Greenland meltwater on future climate suggests that Antarctic meltwater will have a dominant effect on atmospheric cooling and the AIS” confusing.

As I understand, the study by Li et al. (2023) suggests a dominant impact of Antarctic meltwater on the atmospheric cooling in the Southern Hemisphere. The meridional overturning circulation and atmospheric cooling in the Northern Hemisphere is dominated by Greenland meltwater.”

We agree and removed that statement.

“4) Lines 230-233

“... decreasing meridional SST gradients in the South and North Pacific caused by high-latitude warming (Supplementary Fig. 3; Δ SST: U1337 – U1543; U1337 – ODP88213) likely reduced the strength of both the ACC and the PMOC.”

I feel the relation between the PMOC strength and meridional SST gradient in the North Pacific is unclear. For instance, when meridional SST gradient in the North Pacific reached its peak at around 2.7-2.8 Ma, the PMOC strength as indicated by carbonate preservation reached its minimum (Figure S3d and S3f). Moreover, the variations in the PMOC strength hardly exhibit relations to the meridional SST gradient in the North Pacific from 4 Ma to 3 Ma.”

We thank Reviewer #2 and reworded this section because we only refer to long-term 400-kyr cycles. We added the following sentences in lines 291-295: **“It has to be noted that the reconstructions of the SST gradients are based on 100 kyr-binned SST_{UK'37} records hence cannot accurately reflect conditions during glacial-interglacial cycles and should be viewed as long-term trends. This also explains why the records of SST gradients may not completely capture the much higher resolved CaCO₃ records but reflect similar long-term changes.”**

The similar patterns of the North and South Pacific CaCO₃ records, SST gradients, and ACC intensity is now better visible by the smoothed trends (now main Fig. 5). The discussion on the relation between ocean circulation changes and carbonate preservation has been moved to an additional chapter **“Pacific Ocean circulation changes and the carbonate preservation effect”**.

“Minor Comments:

1) Lines 67, 70, 99 and others

The authors use several expressions for million years, such as Myrs and Ma. Please consider using only one abbreviation.”

We use “Ma” for specific time events, whereas “Myrs” is used for time spans (duration).

“2) Line 141

The word “Circulation” seems to be redundant.”

It has been removed.

“3) Lines 161 and 169 “central equatorial Pacific”

I feel it is better to use “eastern equatorial Pacific”, as U1337 is located in the realm of the equatorial Pacific cold tongue.”

In agreement with this comment, it has been changed and “central” has been replaced by “eastern” in lines 195, 338, 816, and 817.

“4) Line 413

It should be reference # 36.”

We replaced the reference number in the list and text accordingly. It is now no. 52.

“5) Line 505 Caption for Figure 3

Lines 540 and 541 Caption for Figure S3

Line 553 Caption for Figure S4

It should be “eastern equatorial Pacific”.”

Done.

Reviewer #3 (Remarks to the Author):

“The paper presents a crucial new record of sea surface temperature in the southeast Pacific over the late Neogene. Combined with previously published grain size data for the Neogene from another site (Lamy et al., 2024), it documents the local ocean temperature change in a crucial sector of the Southern Ocean, where the ACC interacts with geographic obstructions. It documents peculiar gradual shifts in regional SST trends that are decoupled from those in the benthic foraminifer isotope trends. This in itself is a really interesting result, and I congratulate the authors for putting this together. By comparing this new record to some other SST records in the Southern Ocean (although several crucial datasets are unfortunately omitted), delivers a reconstruction of South Pacific temperature and ocean circulation changes, that, according to the authors is fully centered around variations in strength of the ACC. Although the record of sortable silt does express convincingly dramatic changes in bottom ocean flow at this site, I am not convinced of the somewhat unidirectional interpretation that that reflects only strength in ACC flow.

I find the paleotemperature record fascinating, but I have some technical questions regarding the smoothing of the data that I need to be informed of. I will elaborate my concerns below, but in short, the manuscript should:

-put their results in the context of recent insight into the development of the ACC (Evangelinos et al., 2023)

-be complete in the integration of existing SST datasets that cover this time and are from the Southern Ocean, to complete the picture of SST developments

-explain how their interpretation of ocean circulation changes and ACC strength fits into

observations of a DECREASING meridional SST gradient from the subtropics to Antarctica throughout the Neogene (Hou et al., 2023a)

-give proper attention to the fact that not the ACC but ocean fronts reflect meridional SST gradients, and that these shifted in latitude profoundly through geologic time (Hou et al., 2023b). The authors must convincingly show that their reconstructions are indeed a sign of ACC strength rather than latitudinal shifts, and properly explain why their records argue against latitudinal shifts in fronts, or properly explain in the paper why they believe these frontal shifts are not important in explaining the trends they see. Finally they must explain how variations in ACC strength can induce such strong local SST variability as they see in their record.”

We are grateful for the positive assessment of new dataset as its crucial importance for paleoclimatic and oceanographic insights in the Southern Ocean. We thank Reviewer #3 for the thorough review and for carefully suggesting improvements of our manuscript. We now substantially improved the interpretation of our SST record from Site U1543 and the ACC strength record from Site U1541. In particular, we improved the discussion on the important role of oceanic front migrations for both records. We carefully revised the manuscript by strengthening the discussion regarding potential latitudinal shifts of oceanic fronts in relation to changes of reconstructed ACC strength and SST gradients as explained in detail below. We added information on the ACC development since the late Miocene (Evangelinos et al., 2024; Huang et al., 2024). We added the proposed SST records and additional stacked global and Southern Ocean temperature record to the figures and discussion. We refer to the apparently contrasting SST gradients in the Eastern South Pacific and Tasmanian gateway area (Hou et al., 2023a, b). Moreover, we added Southern Oceanic Fronts as well as additional study sites now discussed in the text to the map in the main figure 1.

“Context of other papers on meridional temperature gradient development

Recent efforts have shown the complexity of the developments in the Southern Ocean oceanography, with dating the onset of modern-like ACC flow younger and younger (Evangelinos et al., 2023), in spite of absence of geographic barriers, and with development of the latitudinal temperature gradient in the Southern Ocean contrasting to what one would expect (Hou et al., 2023a). Also previous work has demonstrated the strong mid-latitude cooling in the Southern Ocean, and the effect of polar amplification in the development of the Neogene subtropical gyre (Liu et al., 2022). In light of these papers, none of which are cited in this paper, I am unsure what exactly the innovation of this paper is, although new SST records showing regional SST trends, are always insightful. For instance, the authors mention in the introduction that the westerly winds and ACC strength are strongly governed by the latitudinal temperature gradient. In the discussion I understand that this refers to the meridional temperature gradient of the subtropical gyre, and not that of the Southern Ocean itself. Still, Hou et al., 2023a showed in the compilation of SST records a decrease in latitudinal temperature gradient between the subtropical front and the Antarctic continental margin since the middle Miocene. How do the authors reconcile this with their statements? Is the meridional temperature gradient not set by the position of the fronts which, in turn are governed by the strength and position of the westerly winds? How is the ACC affected by the

fact that the subpolar meridional SST gradient, the one where the ACC is actually in, decreases over the Neogene? Perhaps there is a logical answer to this, but I find this aspect missing in the paper altogether.”

We agree that the presentation and discussion of other published SST records and SST gradients was partly incomplete. In the original supplementary Figure S2, we now added SST records from sites AND-1B/Ross Sea (Duncan et al., 2023), DSDP 594 (SW Pacific; Herbert et al., 2016); ODP 1168/offshore Tasmania (Hou et al., 2023a), ODP 806/western equatorial Pacific (Zhang et al., 2014), and U1337/equatorial Pacific Cold Tongue (Liu et al., 2019). For a better overview and comparability of the individual SST records, we added a new main Figure 3. Since the benthic oxygen isotope record (Westerhold et al., 2020) does not only mirror changes in temperature but also in sea level, we added in the main Figure 2 the stacked global mean surface temperature record as well as the Southern Hemisphere extratropical (30°S-45°S) temperature records since the Pliocene (Clark et al., 2024).

We also added the proposed and another study dating the onset of the modern-like ACC into the text, where it reads in lines 58-61: **“A key component of the MOC is the Antarctic Circumpolar Current (ACC) in the Southern Ocean (SO) with a surface to deep water circumpolar flow that likely evolved during the late Miocene cooling ~10 Ma⁴ approximating its modern circulation characteristics in the South Pacific ~5 Ma⁵ (Fig. 1).**

In lines 73-80, we provide additional information on the ACC development including its past latitudinal movements: **“The development of the modern ACC was likely linked to the intensification North Atlantic deep-water (NADW) production and Ross Sea bottom water production coinciding with the starting Panama seaway closure around 9-10 Ma⁵. The latter’s final closure around 5 Ma was most likely responsible for further intensification of the Atlantic MOC (AMOC) and the ACC⁵. The position of the frontal system of the ACC varied through the past. For instance, there is evidence that the ACC migrated by up to 2° equatorward and up to 6° poleward over the last two deglaciations as reconstructed by sea surface temperature gradients in the southern Indian Ocean¹¹.”**

In addition, we complemented several paragraphs by discussing on the impact of frontal migrations on the ACC and SST gradients. In lines 67-72, we introduced some general aspects of Southern Ocean fronts regarding the morphology of frontal boundaries and their pronounced temperature gradients. We also point to the different latitudinal extent of oceanic fronts in the Southern Ocean and to its past latitudinal migrations over glacial cycles including the ACC.

We are aware that the Zr/Rb based record of the ACC strength is accompanied by changes in the position of ocean fronts and the ACC itself and are reflected by SST gradients, which we now clearly point out throughout the manuscript. We also give attention to the important role of the extent of the Antarctic Ice Sheet to latitudinal migrations of ocean

fronts and shifts in the SWW intensity (lines 63-67, 203-214, 270-276, 335-337, 355-357, 386-389).

In lines 215-220, we further include studies by Hou et al. showing the importance of the equatorial to subtropical SST gradient on Southern Ocean fronts. We also commentate the apparently contradicting long-term SST gradients between i) the equator and subantarctic ESP (increasing; this study) and ii) the subtropics and Antarctica (decreasing; Hou et al., 2023) in lines 220-233. We argue that the apparent discrepancy is not necessarily contradicting. Both SST gradients reflect high latitude cooling in the ESP and in the Tasmanian gateway region. There is currently no SST record from the subtropical Eastern Pacific available to confirm a similar pattern in the Pacific as seen by Hou et al. in the Tasmanian realm. It might be reasonable that the expansion of the Antarctic Ice Sheet, oceanic frontal movements as well as SST gradients developed differently in the different parts of the Southern Ocean.

“Incomplete comparison to other records

It is strange that the high-resolution SST data from Site 594, 806, (Herbert et al., 2016) The record from the Ross Sea (Duncan et al., 2023) and Site 1168 (Hou et al., 2023a Climate of the past) are missing in this paper. Besides those papers paving the way for the inferences made in this paper, they provide crucial clues about the development of meridional temperature gradients and frontal systems (particularly in Hou et al., 2023b nature communications). Even if there are concerns to compare SST records from different proxies, Site 1168 has a convincing comparison between UK37 and TEX86-based SST and thus TEX86 can be used as SST proxy without doubt. Site 594 and those of the western pacific warm pool are from UK37.”

We thank the reviewer for pointing us to additional relevant SST records. We have now added these SST records to original Figure S3 (now main Figure 3) and discuss these records in the main text. We describe differences and similarities in the SST patterns. For example, in lines 171-175, we point to a similar cooling offshore Tasmania (Hou et al., 2023a) and the SW Pacific (Herbert et al., 2016) at 3.7 Ma. In lines 323-331, we refer to additional studies from the Southern Ocean that indicate warming and a reduced Antarctic Ice Sheet during the iNHG.

“ACC strength versus latitudinal shifts

My other main concern is that this paper unidirectionally points to ACC strength variability as a means to explain the data, and completely seems to ignore changes in the latitudinal position of the ACC and the fronts and the local oceanographic effect of those. There is an overwhelming amount of evidence from the modern (e.g., a review in Chapman et al., 2020), Pleistocene (Bard and Rickaby, 2009; Ai et al., 2023) and Neogene (Hou et al., 2023b) that fronts, which determine the meridional temperature gradients much more than the ACC, shift in position, due to changes in the position of the westerlies (Graham et al., 2012). There is another reason to assume more influence of frontal system shifts driving the signals in the technical notes on the SST curve below.”

Thank you again for raising the incomplete discussion on frontal migrations associated with the variability in ACC strength and SST gradients. As mentioned above, we have now included that aspect in much more detail. Thus, in light of modern SST contrasts across Southern Ocean Fronts (Fig. 1), we now argue that frontal movements could explain the partly pronounced SST changes in the Eastern Pacific Ocean.

The recent paper by Lamy et al. (2024) discusses the importance of frontal movement versus overall changes in ACC strength. Based on a cross-frontal sediment core transect in the Southwest Pacific, they find decreased ACC strength during glacials across the fronts (compared to increased currents during interglacials). This does not exclude frontal shifts but points to both overall ACC strength and changes due to the fronts as major factors, at least in the later part of the Pleistocene.

“Technical aspects of the new SST curve

The temporal resolution of the new data is impressively high but the SST data itself is also quite variable from sample to sample, with amplitudes of variability of up to 5 degrees. First of all, I note that the frequency of that variability is not fully captured by the temporal resolution of the data, and this means that the amplitude of variability might be underestimated, which may affect adequate presentation of the trends. Secondly, and this relates to my point made previously, this variability demonstrates that the majority of the oceanographic variability at this site reflects a latitudinal shift in fronts. SST decreases rapidly per latitude across these fronts. As a result, local SST variability is profoundly amplified as the front shifts over the site. Climate forcing alone cannot explain such high variability in the SST. In light of this I do not understand why all emphasis in this paper is put on variability in strength. I am not denying that there was any change in strength of the ACC, I am just surprised how little focus there is in the paper on this factor. If the ACC strength was the only thing that varied, why would SST be so variable? The zonal flow of the ACC would not be able to change local SST just by changing in strength, right? The authors must acknowledge that the amplitude of variability in their SST record reflects ocean frontal shifts, and note that the temporal resolution is unable to fully capture the cyclicity which causes underestimation of the amplitude of variability AND may have some influence on the expression of the trends.”

We thank the reviewer for this relevant comment and the suggestion of a more detailed discussion of oceanic frontal migrations, which is now included in the manuscript. We now clearly relate the enhanced/reduced latitudinal SST gradients to expansion/waning of the Antarctic Ice Sheet that favoured northward/southward movements of oceanic fronts and the ACC as well increased/weakened Southern Westerlies thereby strengthening/weakening the ACC (lines 203-214, 226-229, 270-272, 302-305, 335-337, 355-357, 386-389).

Moreover, we agree that SST amplitudes could potentially be underestimated. We now give credit to this aspect in lines 147-151 mentioning the limitations in e.g. reflecting glacial-interglacial variability and the full SST amplitudes due to the 25-kyr resolution of

our record. In lines 291-295 we also point to the limitations of the 100-kyr-binned SST records in resolving glacial-interglacial cycles.

“Minor points:

Abstract: Abstract does not specify what data is really new in terms of data, or reconstruction”

We revised the respective paragraphs in lines 39-42 clarifying that we present new sea surface temperature (SST) and carbonate preservation data from the subantarctic Eastern South Pacific dating back to the late Miocene, and that we extended the already published 5.3 Ma old record of ACC strength in the Central South Pacific to 8 Ma.

“Main text: “Late Miocene” is not a formal time zone in the Geologic time scale, and thus “Late” should not be capitalized”

We corrected that throughout the manuscript.

“Lines 102/103: I find the explanation of the CaCO₃ records and what they mean in terms of oceanography a bit too simple. For instance: it doesn’t give credit to the fact that surface ocean carbonate production changes strongly in the different ocean zones. The carbonate factory may, in a deep ocean with corrosive bottom waters, play a large role in determining the amount of calcium carbonate preservation. The text and referencing at this part is quite general and not directly geared towards the local processes at hand. I am not convinced that the local calcium carbonate record indeed reflects the amount of global ocean CO₂ outgassing as the authors suggest. At least, the authors have not fully excluded alternative hypothesis that may have driven calcium carbonate preservation at this site.”

We thank you for this comment and agree that the causes for the observed CaCO₃ changes were insufficiently discussed, as also raised by Reviewer#1. Therefore, we now discuss this in more detail in lines 130-143 mentioning that the carbonate content may reflect both surface primary productivity and preservation. At site U1543, preservation/dissolution was reported (Kasuya et al., 2024) playing a major role for glacial-interglacial carbonate variability, whereas productivity plays only a minor role. This is supported by a comparison of sedimentary carbonate with foraminiferal fragmentation ratios and shell weights, with the two latter independent of productivity (Kasuya et al., 2024).

As mentioned above, this topic is now discussed in a separate chapter (“**Pacific Ocean circulation changes and the carbonate preservation effect**”). We agree, that the local CaCO₃ record does not reflect the amount of global ocean CO₂ outgassing. In lines 314-317, we propose that, both, an increased meridional and circumpolar ocean circulation during the Pliocene likely favoured deep-water CO₂ outgassing, with contribution from Eastern South Pacific ventilation.

“Line 113: can community shifts in alkenone producers be excluded? Is there a record of the C₃₈me alkenones to confirm this?”

In lines 153-156 we note: “Possible shifts in alkenone-producing assemblages most likely not biased the U^K₃₇ paleothermometer due to the positive linear correlation between the U^K₃₇ and the U^K_{38Me} indices (methods, Supplementary figures 3, 4).” We added two

supplementary figures showing these indices. According to Zheng et al. (2019) and Guitián et al. (2023), a good correlation between both these indices demonstrates that changes in SST and not in alkenone producer assemblages explain variations in the U^{K}_{37} paleothermometer.

“Line 136: how does this play out in Rohling et al.’s more recent, and more sophisticated deconvolutions of the benthic isotope stack?”

Thank you for drawing our attention to the more recent study by Rohling et al., which does not support a first major glaciation at 2.15 Ma any longer. Therefore, we removed that aspect in the text and in the figures, but point to a later cooling in the southern hemisphere during the iNHG since ~1.8 Ma as demonstrated by the stacked Southern Hemisphere extratropical temperature record by Clark et al. (2024) now shown in Figure 2c.

“Line 220: very speculative. First of all, the Antarctic peninsula never had much ice to melt, which limits the amount of freshwater delivery. Second of all it is unsure how far that meltwater travelled, third of all, it may only locally have reduced CO₂ outgassing, with unknown feedbacks in the rest of the ocean. Plus if there is one “machine that makes the freshening effect of meltwater disappear it is the deep-reaching ACC.”

Thank you for this comment. We understand that our suggestion of the relation between warming, meltwater supply and AMOC weakening is speculative. As comparable concerns came also from Reviewer#1, we rearranged the entire paragraph and added further indications for our proposed impact of the AIS meltwater on AMOC weakening in lines 322-331 and 346-374. In the response letter above on page 2 we argue:” Besides the evidence for a highly dynamic AIS during the iNHG, which was likely related to episodic meltwater release, we now also show other SST records (new main figure 3) in the Southern Ocean (i.e, South Atlantic, Western South Pacific) that clearly show warming during the iNHG, which would suggest general AIS meltwater release also beyond our study area. Therefore, we expect increased transport of freshwater towards the North Atlantic that consequently reduced NADW production and hence hampered AMOC (Stouffer et al., 2007; An et al., 2024). The resulting limited heat supply to the Northern Hemisphere could have favoured the build-up of Northern Hemisphere ice sheets. Finally, we mention a recent modelling study, which demonstrates that meltwater supply, particularly from the ESP, has a pronounced effect on AMOC weakening (An et al., 2024). Despite these more comprehensive data sets, direct proxy evidences for freshening would be needed to further support our theory, but these are not available.“

We would also like to additionally respond to your specific notes:

First, even if the impact of WAIS melting might be overestimated, we refer to other studies indicating warming in the SW Pacific and South Atlantic that, together with warming in the ESP, were potentially related to AIS melting. The latter is indicated not only by simulations, but also by foraminiferal assemblages from the Ross Sea (Seidenstein et al., 2024) as well as by opal and diatom analyses from the Antarctic Peninsula (Cowan et al., 2008), which we now mention in lines 328-331. Therefore, the warming during the iNHG

not only in the ESP, but also in other parts of the Southern Ocean was possibly associated with AIS melting and southward migration of oceanic fronts during this period.

Second, it is possible to assume that at least freshwater in the South Atlantic (derived from the EAIS) contributed to a weakened AMOC, even if only small amounts of freshwater from the South Pacific may have reached the South Atlantic. Nevertheless, in the manuscript we also refer to a recent modelling study suggesting that especially meltwater from the Bellingshausen Sea (i.e., closest to the ESP) may have the largest impact on AMOC weakening (An et al., 2024).

And third, the reduced CO₂ outgassing might have occurred locally in the ESP. However, the “deep-reaching ACC” most likely weakened in strength during that time and could have resulted in increased circumpolar deep ocean CO₂ storage in other parts of the Subantarctic Southern Ocean. We further refer to increased CO₂ storage in the NW Pacific as indicated by reduced CaCO₃ preservation in that region, which might have contributed to iNHG cooling as well. We conclude that the new ESP proxy record provides additional support for a general reduced ocean circulation that likely caused increased CO₂ storage in the deep ocean thereby contributing to the progressive global cooling at the Pliocene-Pleistocene transition.

“Line 113–122: Main point that demonstrates the importance of the position of ocean zones and fronts: if one was to translate these coolings, in comparison to the atmospheric CO₂ decline, to climate sensitivity and polar amplification (somewhere between 2 and 4 per CO₂ doubling and somewhere between 2 and 3 x times global average), one would arrive at ridiculously high numbers none of which are confirmed by model or climate theory. Conclusion: local ocean conditions must have changed such that ocean temperature change was amplified compared to the climate forcing. In a latitudinal belt of the strongest latitudinal temperature gradient, the easiest way to amplify climate change is by latitudinal shifts of ocean zones.”

As developed in the response above, we carefully revised the manuscript by considering this important aspect now in more detail and in light of modern SST contrasts across Southern Ocean Fronts (Fig. 1), we now argue that frontal movements could explain the partly pronounced SST changes in the Eastern Pacific Ocean (lines 206-214; 268-272).

“Line 226: Note that that freshening effect came about by a episodic release from a HUGE ice sheet, in a MUCH smaller ocean basin without an ACC. The analogy to the southern hemisphere is not so clear to me.”

We thank the reviewer for this important comment and agree that our initial sentence might be misinterpreted, because that study (Li et al., 2023) considered indeed meltwater from the Northern Hemisphere. Our intension was to point to a potential role of additional meltwater from the Southern Hemisphere for AMOC. To avoid any misunderstanding, we removed the sentence. Instead, we refer to a recent study by An et al. (2024), which suggests that Antarctic meltwater supply can cause AMOC weakening.

“Ai, X.E., Thöle, L.M., Auderset, A. et al. The southward migration of the Antarctic Circumpolar Current enhanced oceanic degassing of carbon dioxide during the last two deglaciations. *Commun Earth Environ* 5, 58 (2024). <https://doi.org/10.1038/s43247-024-01216-x>

Bard, E., Rickaby, R. Migration of the subtropical front as a modulator of glacial climate. *Nature* 460, 380–383 (2009). <https://doi.org/10.1038/nature08189>

Duncan, B., McKay, R., Levy, R. et al. Climatic and tectonic drivers of late Oligocene Antarctic ice volume. *Nat. Geosci.* 15, 819–825 (2022). <https://doi.org/10.1038/s41561-022-01025-x>

Evangelinos, D., Etourneau, J., van de Flierdt, T. et al. Late Miocene onset of the modern Antarctic Circumpolar Current. *Nat. Geosci.* 17, 165–170 (2024). <https://doi.org/10.1038/s41561-023-01356-3>

Graham, R. M., A. M. deBoer, K. J. Heywood, M. R. Chapman, and D. P. Stevens (2012), Southern Ocean fronts: Controlled by wind or topography?, *J. Geophys. Res.*, 117, C08018, doi:10.1029/2012JC007887.

Herbert, T., Lawrence, K., Tzanova, A. et al. Late Miocene global cooling and the rise of modern ecosystems. *Nature Geosci* 9, 843–847 (2016). <https://doi.org/10.1038/ngeo2813>

Hou, S., Lamprou, F., Hoem, F. S., Hadju, M. R. N., Sangiorgi, F., Peterse, F., and Bijl, P. K.: Lipid-biomarker-based sea surface temperature record offshore Tasmania over the last 23 million years, *Clim. Past*, 19, 787–802, <https://doi.org/10.5194/cp-19-787-2023>, 2023a

Hou, S., Stap, L.B., Paul, R. et al. Reconciling Southern Ocean fronts equatorward migration with minor Antarctic ice volume change during Miocene cooling. *Nat Commun* 14, 7230 (2023b). <https://doi.org/10.1038/s41467-023-43106-4>

Liu, X., Huber, M., Foster, G.L. et al. Persistent high latitude amplification of the Pacific Ocean over the past 10 million years. *Nat Commun* 13, 7310 (2022). <https://doi.org/10.1038/s41467-022-35011-z>

Peter Bijl“

References

An, S.-I., Moon, J.-Y., Dijkstra, H.A., Yang, Y.-M., Song, H. Antarctic meltwater reduces the Atlantic meridional overturning circulation through oceanic freshwater transport and atmospheric teleconnections. *Communications Earth & Environment* 5, 490 (2024). DOI: 10.1038/s43247-024-01670-7

Clark, P.U., Shakun, J.D., Rosenthal, Y., Köhler, P., Bartlein, P.J. Global and regional temperature change over the past 4.5 million years. *Science* 383, 884-890 (2024). DOI: 10.1126/science.adi1908

Cowan, E.A., Hillenbrand, C.-D., Hassler, L.E., Ake, M.T. Coarse-grained terrigenous sediment deposition on continental rise drifts: A record of Plio-Pleistocene glaciation on the Antarctic Peninsula. *Palaeogeography, Palaeoclimatology, Palaeoecology* 265, 275-291 (2008). DOI: 10.1016/j.palaeo.2008.03.010

Duncan, B., McKay, R., Levy, R., Naish, T., Prebble, J.G., Sangiorgi, F., Krishnan, S., Hoem, F., Clowes, C., Dunkley Jones, T., Gasson, E., Kraus, C., Kulhanek, D.K., Meyers, S.R., Moossen, H., Warren, C., Willmott, V., Ventura, G.T., Bendle, J. Climatic and tectonic drivers of late Oligocene Antarctic ice volume. *Nature Geoscience* 15, 819-825 (2022). DOI: 10.1038/s41561-022-01025-x

Evangelinos, D., Etourneau, J., van de Flierdt, T., Crosta, X., Jeandel, C., Flores, J.-A., Harwood, D.M., Valero, L., Ducassou, E., Sauermilch, I., Klocker, A., Cacho, I., Pena, L.D., Kreissig, K., Benoit, M., Belhadj, M., Paredes, E., Garcia-Solsona, E., López-Quirós, A., Salabarnada, A., Escutia, C. Late Miocene onset of the modern Antarctic Circumpolar Current. *Nature Geoscience* 17, 165-170 (2024). DOI: 10.1038/s41561-023-01356-3

- Gutián, J., Stoll, H.M. Evolution of Sea Surface Temperature in the Southern Mid-latitudes From Late Oligocene Through Early Miocene. *Paleoceanography and Paleoclimatology* 36, e2020PA004199 (2021). DOI: 10.1029/2020PA004199
- Haug, G.H., Tiedemann, R. Effect of the formation of the Isthmus of Panama on Atlantic Ocean thermohaline circulation. *Nature* 393, 673-676 (1998). DOI:10.1038/31447
- Herbert, T.D., Lawrence, K.T., Tzanova, A., Peterson, L.C., Caballero-Gill, R., Kelly, C.S. Late Miocene global cooling and the rise of modern ecosystems. *Nature Geoscience* 9, 843-847 (2016). DOI:10.1038/ngeo2813
- Hou, S., Lamprou, F., Hoem, F.S., Hadju, M.R.N., Sangiorgi, F., Peterse, F., Bijl, P.K. Lipid-biomarker-based sea surface temperature record offshore Tasmania over the last 23 million years. *Climate of the Past* 19, 787-802 (2023a). DOI: 10.5194/cp-19-787-2023
- Hou, S., Stap, L.B., Paul, R., Nelissen, M., Hoem, F.S., Ziegler, M., Sluijs, A., Sangiorgi, F., Bijl, P.K. Reconciling Southern Ocean fronts equatorward migration with minor Antarctic ice volume change during Miocene cooling. *Nature Communications* 14, 7230 (2023b). DOI: 10.1038/s41467-023-43106-4
- Huang, H., Gutjahr, M., Song, Z., Fietzke, J., Frank, M., Kuhn, G., Hillenbrand, C.D., Christl, M., Garbe-Schönberg, D., Goepfert, T., Eisenhauer, A. Seawater Lead Isotopes Record Early Miocene to Modern Circulation Dynamics in the Pacific Sector of the Southern Ocean. *Paleoceanography and Paleoclimatology* 39, e2024PA004922 (2024). DOI: 10.1029/2024PA004922
- Kasuya, T., Okazaki, Y., Iwasaki, S., Nagashima, K., Kimoto, K., Lamy, F., Hagemann, J.R., Lembke-Jene, L., Arz, H.W., Murayama, M., Lange, C.B., Harada, N. Orbital timescale CaCO₃ burial and dissolution changes off the Chilean margin in the subantarctic Pacific over the past 140 kyr. *Progress in Earth and Planetary Science* 11, 56 (2024). DOI: 10.1186/s40645-024-00657-4
- Li, Q., Marshall, J., Rye, C.D., Romanou, A., Rind, D., Kelley, M. Global Climate Impacts of Greenland and Antarctic Meltwater: A Comparative Study. *Journal of Climate* 36, 3571-3590 (2023). DOI:10.1175/JCLI-D-22-0433.1
- Liu, J., Tian, J., Liu, Z., Herbert, T.D., Fedorov, A.V., Lyle, M. Eastern equatorial Pacific cold tongue evolution since the late Miocene linked to extratropical climate. *Science Advances* 5, eaau6060 (2019). DOI:10.1126/sciadv.aau6060
- Seidenstein, J.L., Leckie, R.M., McKay, R., De Santis, L., Harwood, D., Scientists, I.E. Pliocene–Pleistocene warm-water incursions and water mass changes on the Ross Sea continental shelf (Antarctica) based on foraminifera from IODP Expedition 374. *J. Micropalaeontol.* 43, 211-238 (2024). DOI: 10.5194/jm-43-211-2024
- Stouffer, R.J., Seidov, D., Haupt, B.J. Climate Response to External Sources of Freshwater: North Atlantic versus the Southern Ocean. *Journal of Climate* 20, 436-448 (2007). DOI: 10.1175/JCLI4015.1
- Westerhold, T., Marwan, N., Drury, A.J., Liebrand, D., Agnini, C., Anagnostou, E., Barnet, J.S.K., Bohaty, S.M., De Vleeschouwer, D., Florindo, F., Frederichs, T., Hodell, D.A., Holbourn, A.E., Kroon, D., Lauretano, V., Littler, K., Lourens, L.J., Lyle, M., Pälike, H., Röhl, U., Tian, J., Wilkens, R.H., Wilson, P.A., Zachos, J.C. An astronomically dated record of Earth's climate and its predictability over the last 66 million years. *Science* 369, 1383 (2020). DOI:10.1126/science.aba6853
- Zhang, Y.G., Pagani, M., Liu, Z. A 12-Million-Year Temperature History of the Tropical Pacific Ocean. *Science* 344, 84-87 (2014). DOI: 10.1126/science.1246172
- Zheng, Y., Heng, P., Conte, M.H., Vachula, R.S., Huang, Y. Systematic chemotaxonomic profiling and novel paleotemperature indices based on alkenones and alkenoates: Potential for disentangling mixed species input. *Organic Geochemistry* 128, 26-41 (2019). DOI: 10.1016/j.orggeochem.2018.12.008

REVISION NOTES: Ref. No.: NCOMMS-24-56160A

The revised parts in the text are here marked in **bold**.

→ Changes in the tracked changes version are highlighted in **yellow**

Reviewer #1 (Remarks to the Author):

“Wegwerth et al. sufficiently addressed many of my former comments in the revised version. However, before possible publication, the authors need to clarify following remaining issues:

The authors now explain much better the possible effects of a freshening in the Southern Ocean on the Atlantic Meridional Overturning Circulation (AMOC) during the iNHG. However, as the authors state that there is yet no proxy evidence for this hypothesized freshening, the whole section dealing with this topic (lines 357-374) is speculative and should be shortened. If the authors could show a proxy record demonstrating significant AIS melting within a region suggested to be critical in affecting the AMOC, this claim would be much stronger.”

First, we would like to thank Reviewer#1 for re-reviewing our manuscript and the general positive assessment of our revision. We agree that the hypothesized freshening appears slightly speculative since we have, unfortunately, no direct proxy evidence for AIS melting during the iNHG. Nevertheless, in the beginning of this chapter, we already referred to several proxy records and model simulations suggesting not only warming but also intermittent retreat of the AIS and sea ice (now lines 320-339).

We shortened the respective paragraph by about 30% and removed statements rather irrelevant for the proposed linkage between AIS meltwater and AMOC weakening in our study and now it reads in lines 359-369: “The transport of **the Antarctic meltwater** from the South towards the North Atlantic by the AMOC could have then reduced NADW production^{56,59}, which would have ultimately weakened the AMOC, reduced the heat supply to the Northern Hemisphere, and favoured the NHG⁴². Although there is currently no proxy evidence for the suggested SO freshening and freshwater forcing on the iNHG due to AIS melting during the long-term SO warming, a recent modelling study on the relation between AIS meltwater and AMOC investigated the role of different locations of meltwater input by hosing experiments in five different sectors of the SO⁵⁹. The authors found that meltwater supply from the West Antarctic marginal seas (i.e., Bellingshausen > Amundsen > Ross Sea), **thus close to Site U1543, would have** the strongest effect on AMOC weakening.”

“The proposed Southern Ocean warming during the iNHG is better elaborated in the revised version. Wegwerth and colleagues now show more SST proxy records from this region. However, the discussion of these SST proxy records still needs to be improved. For example, in lines 324-326 the authors state that SST proxy records from the Southwest Pacific and from the South Atlantic show a warming of about 3 °C during the iNHG. However, the SST records from the Southwest Pacific seen in Figure 3f and g show stable SST or a cooling during 3-2.3 Ma.”

Thank you for enlightening the weakness of the SST description. We carefully revised the respective paragraph by providing more details on the SST development in different parts of the Southern Ocean in lines 320-329: “This **SST increase is largely** consistent with SO warming and Antarctic sea ice retreat **during this time interval as indicated** by proxy data and modelling results^{1,2,13,37,51-52}. **Although interrupted by a cooling, the SST record from the Ross Sea³⁹ shows warmings of up to 6°C at 2.4 Ma and 2.2 Ma (Fig. 3c). Similarly, general warm conditions occurred in the SW Pacific¹ and Tasman Sea¹⁸ with a cooling not before 2.4 Ma (Fig. 3f, g). Even though punctuated by short cooling periods as well, two SST records from the South Atlantic^{1,37} reflect warming around 2.6 Ma and 2.4 Ma (Fig. 3h, i). The high-resolution stacked extratropical Southern Hemisphere temperature record suggest intermittent warming of at least 2.5°C around 2.5 Ma between 30°S and 45°S² (Fig. 2c).”**

“The presented correlation of CaCO₃ preservation in the deep North Pacific and the ACC is very interesting and worth mentioning. However, the discussed influence of the ACC through the Humboldt Current towards the deep North Pacific in lines 302-306 is rather unclear. The model simulation, mentioned in line 51 which is presented to support this link explains that primarily increased salinities in the North Pacific caused an increased Pliocene PMOC and that this led to a strengthened Drake Passage throughflow. I suggest, the authors better describe the connection between the ACC and the North Pacific through the PMOC.”

Thank you very much for indicating the flaw in the discussed relation between the Humboldt Current and deep PMOC. A stronger Humboldt Current would not explain the apparent tight connection between PMOC and ACC. Instead, it is much more conceivable that a strengthened PMOC not only favour an intensified Drake Passage throughflow, but also a strengthened ACC at least in the Pacific, which ultimately controls the Drake Passage throughflow. Therefore, we removed these two sentences (including two references) regarding the originally proposed relation between the Humboldt Current and PMOC and added one sentence (lines 300-310): “The concomitant higher CaCO₃ contents in the high-latitude South and North Pacific Oceans could, therefore, support a strengthened basin-wide PMOC **that coincided with a strengthened ACC**. Model simulations have shown that under Pliocene boundary conditions, an activation of PMOC is associated with a strengthened Drake Passage throughflow⁴⁹, which is ultimately controlled by the ACC strength. **Therefore, an intensified PMOC might not have only favoured an enhanced Drake Passage throughflow, but also a strengthened ACC at least in the Pacific. Thus, since** North Pacific deep water upwells in the SO⁵⁰ and ACC intensities and Pacific convection were concurrently enhanced during parts of the Pliocene^{10,34}, **we suggest a relation between PMOC and ACC on the scale of Pliocene 400 kyr eccentricity cycles¹⁰, similarly to the possible dependence of an enhanced AMOC on the ACC⁵.**”

“Detailed comments:

Lines 63-65: I suggest mentioning the Southern Westerly Winds first.”

It has been rephrased and reads in lines 63-65: “The intensity of the ACC depends on the **strength of the Southern Westerly Winds (SWW)** and the large-scale density structure of the $SO^{3,6-7}$, influenced by the extent of the Antarctic Ice Sheet (AIS) and sea ice.”

“Line 206: “reduced SST” within the ACC?”

Thank you for indicating the missing information. We added that in line 206: “Periods with Pliocene ACC maxima show reduced SST up to about 5°C **in the ESP.**”

“Line 350 and following: What is the effect of meltwater on reconstructed alkenone-derived SST?”

There are several studies suggesting that percentage of $C_{37:4}$ alkenone increases with enhanced freshwater (meltwater) conditions (e.g., Rosell-Melé, 1998; Harada et al., 2008). It might therefore affect the U_{37}^K -based SST because the $C_{37:4}$ alkenone is included in the U_{37}^K -index. Since we use the U_{37}^K -index (without $C_{37:4}$) for SST reconstructions, the effect of meltwater on our reconstructed alkenone-based SST should be negligible.

Reviewer #2 (Remarks to the Author):

“Overall Assessment

The authors have addressed my comments raised in the first round of review and revised the manuscript accordingly. The long-term interplay among the sea surface temperature, $CaCO_3$ preservation, the ACC as indicated by proxies in the Southern Ocean and Southern Hemisphere Westerlies are well explained.

There are a few minor points that I would like to stress upon (the line numbers are from the revised manuscript with track-changes).”

We are grateful for the positive assessment of our revision and thank Reviewer#2 for suggestions improving the manuscript.

“Comments

Lines 75-77

I find this statement confusing. The final closure the Panama Seaway around 5 Ma is indeed responsible for the intensification of the AMOC, yet it weakens the ACC as shown by Figure 7d in Huang et al., (2024). The authors may consider revising this sentence.”

Yes, it was indeed slightly ambiguous. Now it reads in lines 75-77: “The latter’s final closure around 5 Ma was most likely responsible for further intensification of the Atlantic MOC (AMOC) and the **establishment of the modern ACC system**⁵.”

“Lines 170

To avoid confusion, would it be better to move the reference # 36 after the word “record”?”

Yes, thank you. Done.

“Lines 175

The figure citation is incorrect. It should be Fig. 2c, yet not Fig. 1c.”

Thanks for indicating the typo. We corrected the Figure number.

“Line 179

As shown in Fig. 2d, the SSTs increased by approximately 6°C from 3.0 Ma to 2.3 Ma. This sentence may be expressed as something like “SSTUK’37 rose to 12°C reaching ...”

We replaced “by” by “to” in line 179. Thank you for indicating the typo.

“Line 358

To avoid confusion, I prefer replacing “fresher water” with “the Antarctic meltwater?””

Yes, we agree. Done.

“Lines 369-374

I could not agree that the high quantity release of the AIS freshwater to the SO exceeded the modelled threshold. As stated in An et al., (2024), a meltwater input of 1.0 Sv (1 Sverdrup is equivalent to 10^6 m³/s) into the ocean over 100 years accounts for about 10% of the current total Antarctic ice sheet. If the meltwater from the AIS reached the volume threshold, i.e., 0.1 Sv, then the entire AIS would melt after approximately 10,000 years. However, the authors investigate a long-term warming of about 700 kyrs. The quantity of meltwater must be much smaller than the volume threshold.”

Yes, thank you. We agree that the aspect of the volume threshold was not compatible with our original discussion about AIS melting and AMOC weakening. The paragraph was confusing and not helpful in supporting our hypothesis. Since Reviewer#1 recommended shortening this discussion part, we decided to remove it from the manuscript.

Reviewer #3 (Remarks to the Author):

“I have re-reviewed this manuscript after a round of profound revisions. My comments and that of other reviewers have been carefully and dilligently incorporated and I have no further comments to make. The manuscript now reads much better and puts the study in a more complete context in the region.”

We thank Reviewer#3 for re-reviewing and the positive feedback of the revised manuscript and appreciation regarding a broader context of the study.

References:

- Rosell-Melé, A., 1998. Interhemispheric appraisal of the value of alkenone indices as temperature and salinity proxies in high-latitude locations. *Paleoceanography* 13, 694-703.
- Harada, N., Sato, M., Sakamoto, T., 2008. Freshwater impacts recorded in tetraunsaturated alkenones and alkenone sea surface temperatures from the Okhotsk Sea across millennial-scale cycles. *Paleoceanography* 23.

REVISION NOTES: Ref. No.: NCOMMS-24-56160B

The revised parts in the text are here marked in **bold**.

→ Changes in the tracked changes version are highlighted in yellow

Reviewer #1 (Remarks to the Author):

“The authors have considered all my remaining comments, and I do not have more observations.”

We thank Reviewer#1 for re-reviewing the manuscript and are grateful for the final positive assessment.

Reviewer #2 (Remarks to the Author):

“The authors have well addressed most of my concerns. This study is a very good contribution to advance our understanding about the past changes in the ACC and Southern Ocean dynamics.”

We appreciate Reviewer#2 for the general positive feedback of our revised manuscript.

“I still have four some comments for the authors to consider. Yet it appears unnecessary to review the revised manuscript for an additional round.

The line numbers are from the revised manuscript.

1. Lines 64 – 67

“The intensity of the ACC depends on the strength of the Southern Westerly Winds (SWW) and the large-scale density structure of the SO, influenced by the extent of the Antarctic Ice Sheet (AIS) and sea ice. Both influence the position of the frontal system and are controlled by changes in equator-to-pole temperature gradients.”

If I understand correctly, the word “both” here refer to the Southern Ocean westerly and the density structure of the SO, but not the AIS and sea ice. The authors may consider revise the words to avoid confusion.”

Thank you for letting us know about the unfortunate choice of words. We rearranged that paragraph and now it reads in lines 63-67: **“Influenced by the extent of the Antarctic Ice Sheet (AIS) and sea ice, the intensity of the ACC depends on the strength of the Southern Westerly Winds (SWW) and the large-scale density structure of the SO^{3,6-7}. Both affect the position of the frontal system and are controlled by changes in equator-to-pole temperature gradients⁷.”**

“2. Line 96

“... triggers a strengthening ocean circulation and rise of atmospheric CO₂, which in turn results in warming ...”

Would it be better to replace “results in” with “further amplifies”?”

Yes, you are right. It sounds better. We replaced it now in lines 95-96.

“3. Lines 97-99

“... the interplay between climate, ocean circulation, and CO₂ changes, ...”

Would it be better to replace “between” with “among”?”

Yes. Thank you. We replaced it in lines 97-99.

“4. Lines 306-307

“Therefore, an intensified PMOC might not have only favoured an enhanced Drake Passage throughflow, but also a strengthened ACC at least in the Pacific.”

I would change the text to “an intensified PMOC might not only go along an enhanced Drake, ...”.

The intensified PMOC and strengthened ACC are both linked to the freshwater forcing in the simulations. It is difficult to conclude that the enhanced PMOC favours a stronger ACC.”

Thanks. We rephrased the sentence accordingly in lines 307-308: “Therefore, **an intensified PMOC might not only go along an enhanced Drake** Passage throughflow, but also a strengthened ACC at least in the Pacific.”

“5. Data availability

It is better to add an link to the Pangaea repository.””

Yes, we added the DOIs in lines 466-468.

Reviewer #3 (Remarks to the Author):

“I re-reviewed the manuscript, and although I did not have much to comment on in the revisions last time, I do have two minor comments.

In the introduction the authors make some strong connections between ACC strength, Panama isthmus closure and AMOC developments, and although the cited papers do provide some evidence for these connections, I think that the relationships and causalities between these processes are not yet so firmly established. The authors must be careful in stating these causalities, to reflect the ongoing research on these processes.”

We thank Reviewer#3 for re-reviewing the manuscript and suggestions for final improving the manuscript. According to the comment above, we removed “most likely” in the respective sentence and rephrased and softened the statements. Now it reads in lines 73-77: “The development of the modern ACC was likely linked to the intensification **of the** North Atlantic deep-water (NADW) and Ross Sea bottom water production coinciding with the starting Panama seaway closure around 9-10 Ma⁵. The latter’s final closure around 5 Ma **is suggested to be** responsible for further intensification of the Atlantic MOC (AMOC) and the establishment of the modern ACC system⁵.”

“I also have a concern with the new presentation of the SST trends derived from Fig. 3 around the iNHG. I do not see the consistency between the records that the authors propose. The interpretation of warming in other records than the new record of the authors (although also **THAT** warming is not unique in their record) is not clear. There may be sustained warmth in that time interval in other records, though not unique for that time interval. The apparent

assumption that the authors make is that the Southern Ocean should cool or warm uniformly (they see that the warming occurs consistently in all SST records), but it is clearly not uniform and I do not understand why the authors would think that it should be. The Southern Ocean does not have a lot of latitudinal exchange anymore by that time, and so different ocean zones can behave differently, as a result of changes in local conditions or atmospheric heat redistribution. The authors should reflect that better in the inserted part, because I do not necessarily disagree with their review of studies that the AA ice sheet was reduced during the iNHG, or that frontal systems were displaced southwards (Hou et al., 2023 Nature Communications shows that the southward shift of the STF started already by 4 Ma), I am just saying that the manuscript now seems to review the available SST data in search of a uniform warming, and I think that is not required to make that case.”

First, we would like to thank you for raising your concerns regarding the presentation of warming in other Southern Ocean records in Figure 3, which was indeed difficult to identify. Therefore, we added a grey bar to the respective period in the figure, through which warming, intermittent or sustained warmth, is recognizable now in the individual records. Further, we did not state in the manuscript that the iNHG warming is “unique” in the ESP. We just intended to show, that warming, although less pronounced, is seen also in other records. In case, we would not show this, readers might question the ESP warming record because, to our knowledge, it was to date not really shown in the literature, but it was generally rather assumed that cooling during the NHG occurred on a global scale. We hope that our explanation is comprehensible.

Nevertheless, we indeed pointed to a “unique” cooling at 3.7 Ma, which we rephrased in lines 170-172: “This cooling seems to be a prominent feature in the ESP since it is less pronounced in the global benthic $\delta^{18}\text{O}$ record³⁶ or in other Southern and Northern Hemisphere SST_{UK'37} records^{1,2,37}, where SST decreased by no more than 2°C (Figs. 2, 3).” In order to note that temperature changes are not uniform in the Southern Ocean, we added that aspect in lines 175-177: “Similarly, this cooling is not evident in a stacked temperature record from the southern hemisphere extratropics² (30-45°S; Fig. 2c), **which indicates that the Southern Ocean is not necessarily cooling or warming uniformly.**”